# Experimental verification of field-enhanced molecular vibrational scattering at single infrared antennas

Divya Virmani[1], Carlos Maciel-Escudero [1,2], Rainer Hillenbrand [1,3,4] ✉ & Martin Schnell [1,3] ✉

Surface-enhanced infrared absorption (SEIRA) spectroscopy exploits the field enhancement near nanophotonic structures for highly sensitive characterization of (bio)molecules. The vibrational signature observed in SEIRA spectra is typically interpreted as field-enhanced molecular absorption. Here, we study molecular vibrations in the near field of single antennas and show that the vibrational signature can be equally well explained by field-enhanced molecular scattering. Although the infrared scattering cross section of molecules is negligible compared to their absorption cross section, the interference between the molecular-scattered field and the incident field enhances the spectral signature caused by molecular vibrational scattering by 10 orders of magnitude, thus becoming as large as that of field-enhanced molecular absorption. We provide experimental evidence that field-enhanced molecular scattering can be measured, scales in intensity with the fourth power of the local field enhancement and fully explains the vibrational signature in SEIRA spectra in both magnitude and line shape. Our work may open new paths for developing highly sensitive SEIRA sensors that exploit the presented scattering concept.

Infrared (IR) spectroscopy is a widely applied technique for the non-destructive and label-free analysis of materials based on their characteristic vibrational spectra[1]. Owing to the small absorption cross section of molecular vibrations, standard IR spectroscopy typically requires a large quantity of analytes to obtain a measurable signal. This limitation can be overcome by using surface-enhanced IR absorption (SEIRA) spectroscopy, where nanostructures provide strongly confined and enhanced fields at the substrate surface[2,3]. When molecules are placed in a region of strong field enhancement, the spectral signature of molecular vibrations is significantly increased and infrared spectroscopy of small amounts of analytes becomes possible. A higher sensitivity can be achieved by exploiting IR surface plasmon and phonon polariton resonances in nanostructures (antennas) made of metal[4–6] and graphene[7], van-der-Waals material[8] and as well as IR

resonances in dielectric structures[9,10]. These SEIRA architectures allow for increasing the spectral signature by at least five orders of magnitude compared to standard IR spectroscopy, which has helped to push the sensitivity of IR spectroscopy down to the attogram scale[2].

The enhancement mechanism in SEIRA spectroscopy is generally described as the coupling of molecular vibrations with the antenna. The resonant character of this coupling leads to asymmetric spectral line shapes of molecular vibrations in SEIRA spectra, which challenges the direct extraction of molecular information. A variety of theoretical models have been developed to fit and provide an intuitive understanding of the observed line shapes[2,11–15]. Often, vibrational line shapes in SEIRA are described as the Fano-like interference between the electromagnetic field of the (spectrally broad) antenna resonance and the field associated with the (spectrally narrow) molecular vibrations.

[1]CIC nanoGUNE BRTA, 20018 Donostia-San Sebastián, Spain. [2]Materials Physics Center, CSIC-UPV/EHU, 20018 Donostia-San Sebastián, Spain. [3]IKERBASQUE, Basque Foundation for Science, 48013 Bilbao, Spain. [4]Department of Electricity and Electronics, UPV/EHU, 20018 Donostia-San Sebastián, Spain. ✉e-mail: r.hillenbrand@nanogune.eu; schnelloptics@gmail.com

Other models interpret SEIRA in the picture of coupled resonators, where the vibrational line shapes are described as the result of the interference between a bright (spectrally broad) and a dark (spectrally narrow) mode[11] or in the model of two coupled dipoles to predict vibrational line shapes quantitatively and explore the impact of the local field enhancement on the line shapes[15].

So far, it is widely accepted that field-enhanced molecular absorption is the underlying process for the appearance of the enhanced vibrational lines in SEIRA spectra. This idea is rooted in the fact that molecular absorption cross sections are many orders of magnitude larger than molecular scattering cross sections at infrared frequencies. Further, experimental observations show that the spectral signature of molecular vibrations scales with the second power of the local field enhancement, $|f|^2$[16]. On the other hand, Alonso-Gonzalez et al. verified experimentally that the infrared elastic light scattering from a small non-absorbing particle in the vicinity of single metallic infrared antennas is analogous to surface-enhanced Raman spectroscopy (SERS), that is, the intensity of the infrared elastic light scattering from the particle is enhanced by the fourth power of the local field enhancement, $|f|^4$[17]. The underlying mechanism was described as scattering from the particle via the antenna after being illuminated by the antenna[18,19]. Rezus and Selig assumed a small particle exhibiting molecular vibrations in the vicinity of the antenna and showed in a numerical study that field-enhanced molecular scattering fully describes the line shapes in extinction spectra of the antenna-particle system[15]. An experimental verification of field-enhanced molecular scattering and its connection with the molecular vibrational line shapes in SEIRA spectra, however, has been elusive so far.

Here, we explore the role of molecular scattering when molecules with mid-infrared vibrations are located in the vicinity of a single antenna—a system that is usually considered in SEIRA experiments. We develop and numerically verify an analytical model that describes the vibrational line shapes in the extinction cross section as the interference between field-enhanced molecular scattering and the incident field. Performing a near-field experiment, we provide evidence that

field-enhanced molecular scattering can be measured, scales in intensity with the fourth power of the field enhancement provided by the antenna, $|f|^4$, and fully explains the vibrational line shapes in the extinction cross section.

## Results

### Scattering model

We study a typical SEIRA configuration, where a small spherical object (O) exhibiting molecular vibrations is located in the vicinity of an antenna (A) that is tuned to the molecular vibrational resonance (Fig. 1a). Light extinction is detected in the direction of the incident beam, corresponding to a spectroscopy experiment performed in transmission with a transparent substrate. Figure 1b shows the numerically calculated spectral extinction, scattering and absorption cross sections of the antenna-object system, $\sigma^{ext}$, $\sigma^{sca}$ and $\sigma^{abs}$, respectively. We considered a cylindrical gold antenna (A) of 50 nm radius with round ends without a substrate. A spherical nanoparticle (O) with 30 nm radius was placed on the antenna's long axis with a separation of 20 nm between the antenna end and the nanoparticle surface. The molecular vibrational resonance of the nanoparticle material was described by a Lorentzian model, mimicking the Si-CH$_3$ vibration band of PDMS (Polydimethylsiloxane) at 1258 cm$^{-1}$ (for further details see Methods and discussion below). All calculated cross sections clearly show the fundamental dipolar resonance of the antenna as a broad peak (black lines in Fig. 1b), whereas the spectral signature of the molecular vibration appears on top of the antenna resonance, either as a dip or a peak (Fig. 1b, c), illustrating the Fano-type antenna-molecule coupling[20].

In the following, we describe a model for explaining SEIRA spectra based on molecular scattering. The antenna-object system is illuminated by a monochromatic propagating plane wave (Fig. 1a)[15,17],

$$\mathbf{E}^{in}(\mathbf{r}) = \mathbf{E}_0^{in} \exp(i\mathbf{k} \cdot \mathbf{r}), \tag{1}$$

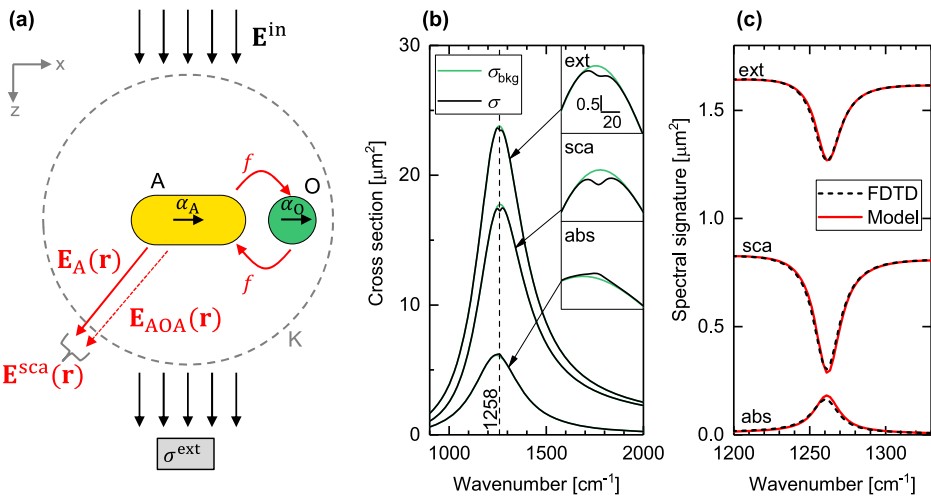

**Fig. 1 | Surface-enhanced infrared vibrational spectroscopy modeled as a scattering process. a** Elastic light scattering of a small particle of a material exhibiting molecular vibrations (O) in the presence of an infrared-resonant antenna (A). **b** Numerically calculated extinction (ext), scattering (sca) and absorption (abs) cross section, $\sigma$ (black lines), of the system in (**a**), where a small particle (sphere of 30 nm radius) was placed at 20 nm distance from the end of an Au antenna (50 nm radius, 3.19 µm length, no substrate). The antenna resonance was tuned to the 1258 cm$^{-1}$ molecular vibrational resonance of the small particle (vertical black dashed line in (**b**)). For reference, the case of a non-absorbing small particle is shown ($\sigma_{bkg}$, green line). **c** Baseline-corrected spectra, as obtained from (**b**), $\Delta\sigma = \sigma - \sigma_{bkg}$, show the spectral signature of the molecular vibrations (black

dashed line), which is fully explained by the scattering model in (**a**) ((Eq. 13), red line). $\mathbf{E}^{in}$: Incident field. $\mathbf{E}_A$: Direct antenna scattering. $\mathbf{E}_{AOA}$: Field-enhanced molecular scattering. $\mathbf{E}^{sca}$: Total scattered field. $\sigma^{ext}$: Light extinction measured by the detector. $\alpha_A$ and $\alpha_O$: Point polarizabilities describing the antenna and small particle, respectively. $f$: Field enhancement provided by the antenna. K: Sphere surrounding the antenna and the small particle. The molecular vibration of the particle material was modeled by a Lorentzian dielectric function described by Eq. (35) with background permittivity $\varepsilon_{bkg} = 1.55$, and resonance $\omega_0 = 1258$ cm$^{-1}$, oscillator strength $\varepsilon_{Lorentz} = 0.1$ and linewidth $\gamma = 10$ cm$^{-1}$ to mimic the Si-CH$_3$ vibration band of PDMS (black lines in (**b**)), while $\varepsilon_{Lorentz} = 0$ was assumed in Eq. (35) for the non-absorbing particle (green lines in (**b**)).

where $\mathbf{E}_0^{in} = E_0^{in}\hat{\mathbf{x}}$ is a constant vector describing the amplitude and polarization ($x$-direction) of the plane wave, which propagates along the direction $\hat{\mathbf{n}} = \mathbf{k}/k$ ($z$-direction) with a wavenumber $k$. The scattered field of the antenna-object system, $\mathbf{E}^{sca}$, may be described as an interaction series that is truncated in the limit of a weakly scattering object[18,19]:

$$\mathbf{E}^{sca}(\mathbf{r}) = \mathbf{E}_A(\mathbf{r}) + \mathbf{E}_O(\mathbf{r}) + \mathbf{E}_{OA}(\mathbf{r}) + \mathbf{E}_{AO}(\mathbf{r}) + \mathbf{E}_{AOA}(\mathbf{r}) + \cdots. \quad (2)$$

In the case of elastic light scattering of a small object in the infrared spectral range, the scattered field, $\mathbf{E}^{sca}(\mathbf{r})$, and the scattered field intensity, $\left|\mathbf{E}^{sca}(\mathbf{r})\right|^2$, can be approximated by the following expressions for large field enhancement factors, $|f| \gg 1$[17]:

$$\mathbf{E}^{sca}(\mathbf{r}) \approx \mathbf{E}_A(\mathbf{r}) + \mathbf{E}_{AOA}(\mathbf{r}), \quad (3)$$

$$\left|\mathbf{E}^{sca}(\mathbf{r})\right|^2 \approx \mathbf{E}_A^*(\mathbf{r}) \cdot \mathbf{E}_A(\mathbf{r}) + 2\text{Re}\left\{\mathbf{E}_A^*(\mathbf{r}) \cdot \mathbf{E}_{AOA}(\mathbf{r})\right\}. \quad (4)$$

For simplicity, antenna (A) and object (O) are approximated as point dipoles with polarizability, $\alpha_A$ and $\alpha_O$, so that

$$\mathbf{E}_A(\mathbf{r}) = k^2 \mathbf{G}_x(\mathbf{r}, \mathbf{r}_A) \alpha_A E_x^{in}(\mathbf{r}_A), \quad (5)$$

$$\mathbf{E}_{AOA}(\mathbf{r}) = k^2 \mathbf{G}_x(\mathbf{r}, \mathbf{r}_A) f \alpha_O f E_x^{in}(\mathbf{r}_A), \quad (6)$$

where $\mathbf{G}_x(\mathbf{r}, \mathbf{r}_A)$ is the $x$-component of the free-space Green's tensor function and $\mathbf{r}_A$ is the position of the antenna. $f$ is the local field enhancement provided by the antenna at the position of the object. Note that $f$ is generally a complex quantity, i.e., $f = |f|e^{i\text{Arg}\{f\}}$, because of the resonant character of the antenna response. The index A in $\mathbf{E}_A$ indicates the direct antenna scattering, that is, the field radiated directly by the antenna after being illuminated by the incident field, $\mathbf{E}^{in}$. The index AOA in $\mathbf{E}_{AOA}$ refers to the double scattering process between antenna and object, where the incident field, $\mathbf{E}^{in}$, excites the antenna, the antenna polarizes the object and the object scatters via the antenna to the far field. The AOA term carries information on the molecular vibration and is termed the field-enhanced molecular scattering. Note that terms to second or higher order in O can be neglected owing to the assumption of small object polarizability, $\alpha_O$. Having obtained an expression for the scattered field, $\mathbf{E}^{sca}$, the extinction cross section may be derived by applying the optical theorem[21],

$$\sigma^{ext} = \frac{4\pi}{k\left|\mathbf{E}_0^{in}\right|^2} \text{Im}\left\{\mathbf{E}_0^{in*} \cdot \mathbf{E}_0^{sca}(\hat{\mathbf{n}})\right\}, \quad (7)$$

which relates the extinction cross section, $\sigma^{ext}$, of the antenna-object system to the scattering amplitude, $\mathbf{E}_0^{sca}(\hat{\mathbf{n}})$, defined as $\mathbf{E}^{sca}(\mathbf{r}) = \frac{\exp(ikr)}{r}\mathbf{E}_0^{sca}(\hat{\mathbf{r}})$ and evaluated in the propagation direction of the incident plane wave, $\hat{\mathbf{n}}$[22]. The scattering cross section may be obtained by integration of the scattered field intensity, $\left|\mathbf{E}^{sca}(\mathbf{r})\right|^2$, over the surface of a sphere K surrounding the antenna-object system[19,23]

$$\sigma^{sca} = \frac{1}{\left|\mathbf{E}_0^{in}\right|^2} \int_K dK \left|\mathbf{E}^{sca}(\mathbf{r})\right|^2. \quad (8)$$

With Eqs. (3), (4), the extinction and scattering cross sections in Eqs. (7) and (8) approximate to

$$\sigma^{ext} \approx \frac{4\pi}{k\left|\mathbf{E}_0^{in}\right|^2} \text{Im}\left\{\mathbf{E}_0^{in*} \cdot \mathbf{E}_{A,0}(\hat{\mathbf{n}}) + \mathbf{E}_0^{in*} \cdot \mathbf{E}_{AOA,0}(\hat{\mathbf{n}})\right\}, \quad (9)$$

$$\sigma^{sca} \approx \frac{1}{\left|\mathbf{E}_0^{in}\right|^2} \int_K dK \left(\mathbf{E}_A^*(\mathbf{r}) \cdot \mathbf{E}_A(\mathbf{r}) + 2\text{Re}\left\{\mathbf{E}_A^*(\mathbf{r}) \cdot \mathbf{E}_{AOA}(\mathbf{r})\right\}\right), \quad (10)$$

where $\mathbf{E}_{A,0}$ and $\mathbf{E}_{AOA,0}$ are the scattering amplitudes of the fields $\mathbf{E}_A(\mathbf{r})$ and $\mathbf{E}_{AOA}(\mathbf{r})$, respectively. Equation (9) provides a description of the extinction cross-section of the antenna-molecule system that is exclusively based on the incident field, $\mathbf{E}^{in}$, the field associated with direct antenna scattering, $\mathbf{E}_A$, and the field associated with field-enhanced molecular scattering, $\mathbf{E}_{AOA}$. Molecular absorption does not explicitly appear as a term in Eq. (9), which is expected because the optical theorem is based exclusively on evaluating the scattered field. As we will discuss below in more detail, the interference of the incident field, $\mathbf{E}^{in}$, with the direct antenna scattering, $\mathbf{E}_A$ (the 1st term in Eq. (9)), describes the antenna resonance in the extinction cross section (Fig. 1b, green line). The interference of the incident field, $\mathbf{E}^{in}$, with the field-enhanced molecular scattering, $\mathbf{E}_{AOA}$ (the 2nd term in Eq. (9)), describes the spectral signature of the molecular vibrations of the object material in the extinction cross section (Fig. 1c). Regarding the scattering cross section (Eq. (10)), the spectral signature of the molecular vibration is described by the interference of field-enhanced molecular scattering, $\mathbf{E}_{AOA}$, with the direct antenna scattering, $\mathbf{E}_A$ (the 2nd term in Eq. (10)), rather than the incident field, explaining the differences in the magnitude and shape of the vibrational line shapes between the extinction and scattering cross section (Fig. 1c, Supplementary Fig. 2). We obtain simple expressions for the extinction and scattering cross sections by inserting Eqs. (5), (6) in Eqs. (9), (10):

$$\sigma^{ext} = k\text{Im}\{\alpha_A + f^2\alpha_O\}, \quad (11)$$

$$\sigma^{sca} = \frac{k^4}{6\pi}\left[|\alpha_A|^2 + 2\text{Re}\left\{\alpha_A^* f^2\alpha_O\right\}\right]. \quad (12)$$

We identify the 1st term in Eqs. (11), (12) as the extinction and scattering cross sections of the unloaded antenna, $\sigma_A^{ext} = k\text{Im}\{\alpha_A\}$ and $\sigma_A^{sca} = \frac{k^4}{6\pi}|\alpha_A|^2$, respectively. The 2nd term in Eqs. (11), (12) is the spectral signature of the molecular vibration in the extinction and scattering cross section,

$$\sigma_{vib}^{ext} = k\text{Im}\left\{f^2\alpha_O\right\}, \quad \sigma_{vib}^{sca} = \frac{k^4}{3\pi}\text{Re}\left\{\alpha_A^* f^2\alpha_O\right\}. \quad (13)$$

Note that ref. 15 arrived at the same expression for $\sigma_{vib}^{ext}$ based on a model of two coupled point dipoles. Further note that the absorption cross section and the corresponding spectral signature of the molecular vibration can be directly obtained by applying energy conservation, $\sigma_{vib}^{abs} = \sigma_{vib}^{ext} - \sigma_{vib}^{sca}$. Equation (13) provides an intuitive explanation for the vibrational line shapes: the apparent polarizability of the object is increased by the square of the field enhancement, $f$, provided by the antenna. Note that the field enhancement, $f$, is generally a complex quantity and shifts the phase of the apparent object polarizability, $f^2\alpha_O$. When taking the imaginary part in Eq. (13), the spectral signature in the extinction cross section, $\sigma_{vib}^{ext}$, thus yields different line shapes depending on the tuning of the antenna with respect to the molecular vibration[15]. In the case where the antenna resonance is tuned to the molecular vibration (that is, $\text{Arg}(f) = \text{Arg}(\alpha_A) = \pi/2$ because of $f = k^2 G_{xx}(\mathbf{r}_O, \mathbf{r}_A)\alpha_A$), we find that Eq. (13) becomes

$$\sigma_{vib}^{ext} = -k|f|^2\text{Im}\{\alpha_O\}, \quad \sigma_{vib}^{sca} = -\frac{k^4}{3\pi}|\alpha_A||f|^2\text{Im}\{\alpha_O\}. \quad (14)$$

Thus, the spectral signature of the molecular vibration in the extinction cross section, $\sigma_{vib}^{ext}$, – based on the scattering model developed above – is formally identical to field-enhanced molecular

absorption, $k|f|^2 \text{Im}\{\alpha_O\}$, when the antenna is tuned to the molecular vibration. Therefore, the vibrational signature in typical SEIRA spectra of single resonant antennas can be equivalently well understood as either field-enhanced molecular absorption or interferometrically- and field-enhanced molecular scattering.

## Interferometric enhancement

It may seem counterintuitive to attribute the spectral signature of molecular vibrations in the extinction cross section to molecular scattering, as illustrated with the following example. Considering an isolated spherical nanoparticle with a radius of 1 nm and a vibrational resonance at 1258 cm⁻¹, the scattering cross section, $\frac{k^4}{6\pi}|\alpha_O|^2 \sim 2.3 \cdot 10^{-31}$ m² is ten orders of magnitude smaller than the absorption cross section, $k\text{Im}\{\alpha_O\} \sim 2.2 \cdot 10^{-21}$ m², and thus negligible (Methods, Supplementary Table 1). Even if the nanoparticle is placed next to an antenna as in Fig. 1a, the field enhancement provided by the antenna ($|f| \sim 50$) cannot compensate for this large discrepancy, that is, the field-enhanced molecular scattering cross section, $\frac{k^4}{6\pi}|f|^4|\alpha_O|^2 \sim 1.7 \cdot 10^{-24}$ m², is still six orders of magnitudes smaller than the field-enhanced molecular absorption cross section, $k|f|^2\text{Im}\{\alpha_O\} \sim 6.1 \cdot 10^{-18}$ m². We point out that these considerations apply to intensity measurements of pure field-enhanced molecular scattering and absorption. However, according to the presented model, field-enhanced molecular scattering, $\mathbf{E}_{AOA}$, interferes with the incident field, $\mathbf{E}^{in}$ (for the extinction cross section, Eq. (9)), or the direct antenna scattering, $\mathbf{E}_A$ (for the scattering cross section, Eq. (10)), yielding an extraordinarily large enhancement of molecular scattering by a factor of

$$F^{ext} = \frac{|\sigma_{vib}^{ext}|}{\sigma_O^{sca}} \approx |f|^2 \frac{6\pi}{k^3|\alpha_O|} \approx 2.9 \cdot 10^{13}, \quad (15)$$

$$F^{sca} = \frac{|\sigma_{vib}^{sca}|}{\sigma_O^{sca}} \approx |f|^2 \frac{2|\alpha_A|}{|\alpha_O|} \approx 4.5 \cdot 10^{13}, \quad (16)$$

respectively. The enhancement factors, $F^{ext}$ and $F^{sca}$, are defined as the ratio of the spectral signature of the molecular vibration in extinction and scattering cross section, $\sigma_{vib}^{ext}$ and $\sigma_{vib}^{sca}$ (Eq. (14)), respectively, to the molecular scattering cross section of the isolated particle, $\sigma_O^{sca} = \frac{k^4}{6\pi}|\alpha_O|^2$. The interferometric enhancement is $6\pi/k^3|\alpha_O| \approx 1.0 \cdot 10^{10}$ in case of the extinction cross section (Eq. (15)), and $2|\alpha_A|/|\alpha_O| \sim 1.6 \cdot 10^{10}$ in case of the scattering cross section (Eq. (16)). In the case of the scattering cross section, the dual role of the antenna becomes apparent, providing both field enhancement and interferometric enhancement of molecular scattering. More precisely, the direct antenna scattering, $\mathbf{E}_A$, can be interpreted as an internal reference field to $\mathbf{E}_{AOA}$ (Eq. (10)) and may become very large compared to $\mathbf{E}_{AOA}$ owing to the much larger polarizability of the antenna, $\alpha_A$, compared to the object polarizability, $\alpha_O$. Thus, interferometric enhancement together with field enhancement ($|f|^2 \sim 2.7 \cdot 10^3$) provides a total enhancement of molecular scattering by 13 orders of magnitude, overcoming the large discrepancy between pure molecular scattering and absorption cross sections.

## Numerical validation

To corroborate the presented scattering model, we compared the numerically calculated spectral signature of the molecular vibration (Fig. 1c) against the analytical expressions of $\sigma_{vib}^{ext}$ and $\sigma_{vib}^{sca}$ from Eq. (13). In the case of an antenna tuned to the molecular vibration at 1258 cm⁻¹, the spectral signature of the molecular vibration is clearly recognizable in form of a dip in both extinction and scattering cross section[20]. The scattering model (red line in Fig. 1c) fully reproduced the numerically calculated spectral signature, $\Delta\sigma^{ext}(\omega)$ and $\Delta\sigma^{sca}(\omega)$ (black dashed line), in both magnitude and peak shape. The deviation of the

scattering model to the numerical calculation was found to be 4.4% and 2.3% for the extinction and scattering cross section, respectively, which we attribute to numerical error (i.e., the challenge to accurately model a very small object (O) next to a large antenna). We further verified the scattering model for the following cases. First, the model yields very good agreement for a range of object sizes (Supplementary Fig. 1). Second, the model accurately reproduced the asymmetric (Fano-like) line shapes of SEIRA spectra. In detail, we considered a fixed molecular resonance and antennas of different lengths to yield antenna resonances either above, at or below the molecular resonance at $\omega_0 = 1258$ cm⁻¹ (Supplementary Fig. 2). Briefly, antennas tuned to above the molecular resonance ($\text{Arg}(f(\omega_0)) \to 0$) yield a spectral signature of the molecular vibration in form of a peak in the extinction cross section ($\text{Im}\{\alpha_O(\omega)\}$). Antennas tuned to the molecular resonance ($\text{Arg}(f(\omega_0)) = \pi/2$) yield a dip ($-\text{Im}\{\alpha_O(\omega)\}$). Antennas tuned to below the molecular resonance ($\text{Arg}(f(\omega_0)) \to \pi$) yield again a peak ($\text{Im}\{\alpha_O(\omega)\}$). Because of the different interference process (cf. Equations (9), (10)), the vibrational line shape in the scattering cross section follows the sequence from normal dispersive ($\text{Re}\{\alpha_O(\omega)\}$) to a dip ($-\text{Im}\{\alpha_O(\omega)\}$) to anomalous dispersive ($-\text{Re}\{\alpha_O(\omega)\}$) as the antenna resonance is tuned to frequencies above, at and below the molecular resonance, $\omega_0$. Third, the model reproduced accurately the coupling of the antenna with weak oscillators (molecular vibrations) but showed significant deviation in case of strong oscillators (phononic-like), which is expected because the assumption of weak coupling is built into the scattering model (Supplementary Fig. 3).

## Experimental verification of field-enhanced molecular scattering

To support the presented scattering model experimentally, and to provide evidence that the interaction between antenna and molecular vibration can indeed be described as a scattering process, we designed an experiment that allowed us to directly measure the spectral signature of the field-enhanced molecular scattering, $\mathbf{E}_{AOA}$, free of any interference with the incident field, $\mathbf{E}^{in}$, or direct antenna scattering, $\mathbf{E}_A$. To this end, a single IR-resonant Au nanorod antenna was illuminated from below through an IR-transparent substrate with a broadband mid-IR laser beam and with the polarization of the incident field, $\mathbf{E}_{in}$, aligned parallel to the nanorod antenna long axis (Fig. 2). A molecule-coated atomic force microscopy (AFM) tip was brought into the near field of the antenna, where the molecules on the AFM tip mimic the molecules in a traditional SEIRA spectroscopy experiment. The AFM tip was vertically vibrated at a frequency $\Omega$ to modulate the distance between the molecules and the nanorod antenna, $d(\Omega)$. Such distance modulation varies the field enhancement, $f$, provided by the antenna at the location of the molecules and thus only affects the field-enhanced molecular scattering, $\mathbf{E}_{AOA}$, while the direct antenna scattering, $\mathbf{E}_A$, is not affected (cf. Equations (5),(6))[17]. The backscattered light from the antenna-tip system, $\mathbf{E}_{sca} = \mathbf{E}_A + \mathbf{E}_{AOA}$, was collected from

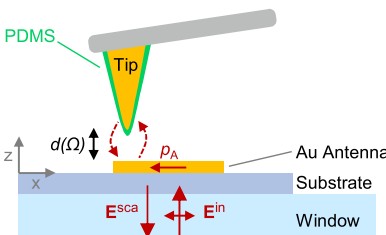

**Fig. 2 | Near-field experiment to measure field-enhanced molecular scattering.** $\mathbf{E}^{in}$: Incident field polarized along the antenna long axis, $p_A$: Induced dipole moment in the antenna. $\mathbf{E}^{sca}$: Scattered field, PDMS: Polydimethylsiloxane. Dashed lines illustrate the near-field coupling between antenna and tip, which is modulated by varying sinusoidally the tip-antenna distance, $d(\Omega)$, at frequency $\Omega$.

below the sample and analyzed with a Michelson interferometer, where the antenna-tip system was placed in one of the interferometer arms[24]. Demodulation of the detector signal at frequency $n\Omega$ and recording the demodulated signal as a function of the reference mirror position yields an interferogram, which was processed by performing Fourier transforms to obtain a complex-valued spectrum $\tilde{s}_n(\omega)$ with amplitude $s_n(\omega)$ and phase $\varphi_n(\omega)$:

$$\tilde{s}_n(\omega) = s_n(\omega)\, e^{i\varphi_n(\omega)} \propto E_{\mathrm{AOA}}(\omega). \tag{17}$$

Importantly, the combination of signal demodulation and interferometric detection allows for suppression of the direct antenna scattering, $\mathbf{E}_A$, and it can be shown that the resulting spectrum, $\tilde{s}_n(\omega)$, is proportional to field-enhanced molecular scattering, $\mathbf{E}_{\mathrm{AOA}}$, in case of the IR-resonant nanorod antennas considered here[17]. Note that illumination and detection directions were identical. As a consequence of reciprocity, the incident field illuminating the object and the field scattered by the object are enhanced by the same factor $f$, thus ensuring $E_{\mathrm{AOA}} \propto f^2$[17].

Figure 3a shows the measured near-field amplitude spectra, $s_n(\omega)$, obtained with rectangular-shaped single gold nanorods (60 nm height, 250 nm width, CaF$_2$ substrate) that were fabricated with lengths ranging from 1.5 μm to 3.5 μm to yield a fundamental dipolar resonance in the mid-IR spectral range. Specifically, by changing the nanorod length, $L$, we could control the antenna resonance with respect to the 1258 cm$^{-1}$ (Si-CH$_3$) vibration of the PDMS coating on the AFM tip. The following three cases were considered: antenna resonance above 1258 cm$^{-1}$ (red line), near 1258 cm$^{-1}$ (blue) and below 1258 cm$^{-1}$ (green). The near-field amplitude spectra (solid lines in Fig. 3a) show the typical fundamental dipolar resonance in form of a broad peak, which shifts to higher frequencies with increasing antenna length, $L$[25]. The spectral signature of the vibrational resonances of PDMS is clearly seen at the marked frequencies (arrows in Fig. 3a). Interestingly, the observed spectral signature is different compared to typical SEIRA line shapes. Specifically, a dispersive line shape (rather than a dip) was observed when the antenna is tuned to the molecular resonance (blue line, $L = 2.6\mu m$) and a step-like shape (rather than a dispersive line shape) in the below-resonance case (green line, $L = 3.0\mu m$).

To corroborate the experimental data, we performed numerical calculations describing the near-field experiment based on finite-difference time-domain (FDTD) calculations[26,27] (Fig. 3b). These calculations describe (i) the near-field coupling between the antenna and a core-shell nanoparticle (50 nm Au core and 10 nm thick PDMS coating) mimicking the molecule-coated AFM tip, (ii) the optical apparatus for collection and refocusing of the scattered light from the sample, and (iii) signal demodulation and interferometric detection, thus providing a comprehensive description of the experimentally recorded data (see Supplementary Note 1 for further details). The numerical calculations reproduced qualitatively the experimental data, and particularly, the shape of the spectral signature of the molecular vibrations was correctly reproduced (cf. solid lines in Fig. 3b, a).

We isolated the spectral signature of the molecular vibrations in both experiment and numerical calculation by (i) calculating the near-field amplitude spectra assuming a core-shell nanoparticle with a non-absorbing dielectric shell that mimics an AFM tip that is coated with a non-absorbing molecular film, $s_3^{\mathrm{bkg}}(\omega)$ (shaded areas in Fig. 3a, b), and (ii) subtracting $s_3^{\mathrm{bkg}}(\omega)$ from the experimental and the numerically calculated scattering spectra, $\Delta s_3(\omega) = s_3(\omega) - s_3^{\mathrm{bkg}}(\omega)$ (see Supplementary Note 1). We found that the spectral signature of the molecular vibration is dispersive for all considered rod lengths (Fig. 3c, d) and resembles the magnitude of the polarizability, $|\alpha_O(\omega)|$, of the core-shell nanoparticle (black solid line in Fig. 3d). This observation provides clear evidence that the interaction between antenna and molecular vibration can be described as a scattering process, and that field-

enhanced molecular scattering can be measured and is a significant quantity.

To relate the spectral signature of the Si-CH$_3$ molecular vibration in Fig. 3c, d with the local field enhancement of the antenna, we extracted the peak-to-peak value of the spectral signature of the molecular vibration, $\Delta s_3^{\mathrm{pp}}$, from the experimental data (Fig. 3c) as a function of nanorod length, $L$, and compared it against the calculated field enhancement, $|f|$. Figure 3e shows a parametric representation of the $\log \Delta s_3^{\mathrm{pp}}(L)$ and $\log|f(L)|$. We found a nearly linear behavior, revealing that the spectral signature, $\Delta s_3^{\mathrm{pp}}$, and field enhancement, $|f|$, are related by a power law (red points in Fig. 3e). Performing a linear least-square fitting, we obtained a nearly quadratic scaling of the spectral signature with $\Delta s_3^{\mathrm{pp}} \propto |f|^{1.8 \pm 0.4}$, which is close to the expected power-of-two law of field-enhanced molecular scattering, $|\mathbf{E}_{\mathrm{AOA}}| \propto |f|^2$, as described in Eq. (18). The corresponding intensity of the spectral signature thus scales with $(\Delta s_3^{\mathrm{pp}})^2 \propto |f|^{3.60}$, confirming that the intensity of field-enhanced molecular scattering scales with the fourth power of the local field enhancement. We obtained a similar scaling law of $\Delta s_3^{\mathrm{pp}} \propto |f|^{1.94 \pm 0.09}$ with the data of the numerical model (obtained from Fig. 3d and plotted as black points in Fig. 3e), which further supports our experimental data. The discrepancy between the experimental and numerical data can be attributed to experimental noise, which rendered measurements of antennas that were far off resonance with the molecular vibration (small $|f|$) less reliable.

In addition, the near-field phase spectra, $\varphi_3(\omega)$, in Fig. 3f, g show the typical response of a field-enhanced molecular scattering process with a transition from 0° to 360° as the fundamental resonance of the antenna is crossed. More precisely, the typical response of an antenna (phase transition from 0° to 180° across the antenna resonance) is doubled owing to the double-scattering mechanism described by the AOA term[17]. The spectral signature of the molecular vibration appears in form of a positive peak on top of the slowly varying phase response of the antenna. By extracting the baseline-corrected phase spectra, $\Delta\varphi_3(\omega) = \varphi_3(\omega) - \varphi_3^{\mathrm{bkg}}(\omega)$, we found that the spectral signature of the molecular vibration resembles the argument of the polarizability, $\mathrm{Arg}\{\alpha_O(\omega)\}$, (Fig. 3h, i). We further found that the peak height, $\Delta\varphi_3^{\mathrm{pp}}$, of the Si-CH$_3$ molecular vibration was approximately independent of the local field enhancement, $|f|$, with a value of about 21° (data points in Fig. 3j), which agrees well with the corresponding peak height in the argument of the polarizability, $\mathrm{Arg}\{\alpha_O(\omega)\}$ (dashed line in Fig. 3j), and is consistent with the phase of the field-enhanced molecular scattering as described in Eq. (19).

To corroborate the presented scattering model, we compared the numerically calculated near-field spectra with the scattering model for SEIRA spectroscopy developed above (dashed lines in Fig. 3b, g). To this end, we calculated the field-enhanced molecular scattering, $\mathbf{E}_{\mathrm{AOA}}$, in Eq. (6) in amplitude and phase:

$$|E_{\mathrm{AOA}}| \propto k^2 |f|^2 |\alpha_O|, \tag{18}$$

$$\varphi_{\mathrm{AOA}} = 2\,\mathrm{Arg}\{f\} + \mathrm{Arg}\{\alpha_O\}, \tag{19}$$

where field enhancement, $f$, was obtained numerically and the polarizability, $\alpha_O$, of the core-shell nanoparticle in Fig. 3b was evaluated analytically (Supplementary Note 1). Note that in contrast to Fig. 1, where the particle was located on the antenna's long axis, here, the particle is located above the antenna's end as in the experiment (schematic in Fig. 3b). Remarkably, the scattering model (dashed lines in Fig. 3b, g) reproduces the main spectral features including that a dispersive line shape of the molecular vibration is observed for all three antenna lengths (solid lines in Fig. 3b,g), confirming that the field-enhanced molecular scattering, $\mathbf{E}_{\mathrm{AOA}}$ (Eq. (6)), is observed.

In the following, we establish with Fig. 4 a connection between the near-field experiment shown in Fig. 2 with the vibrational line shapes

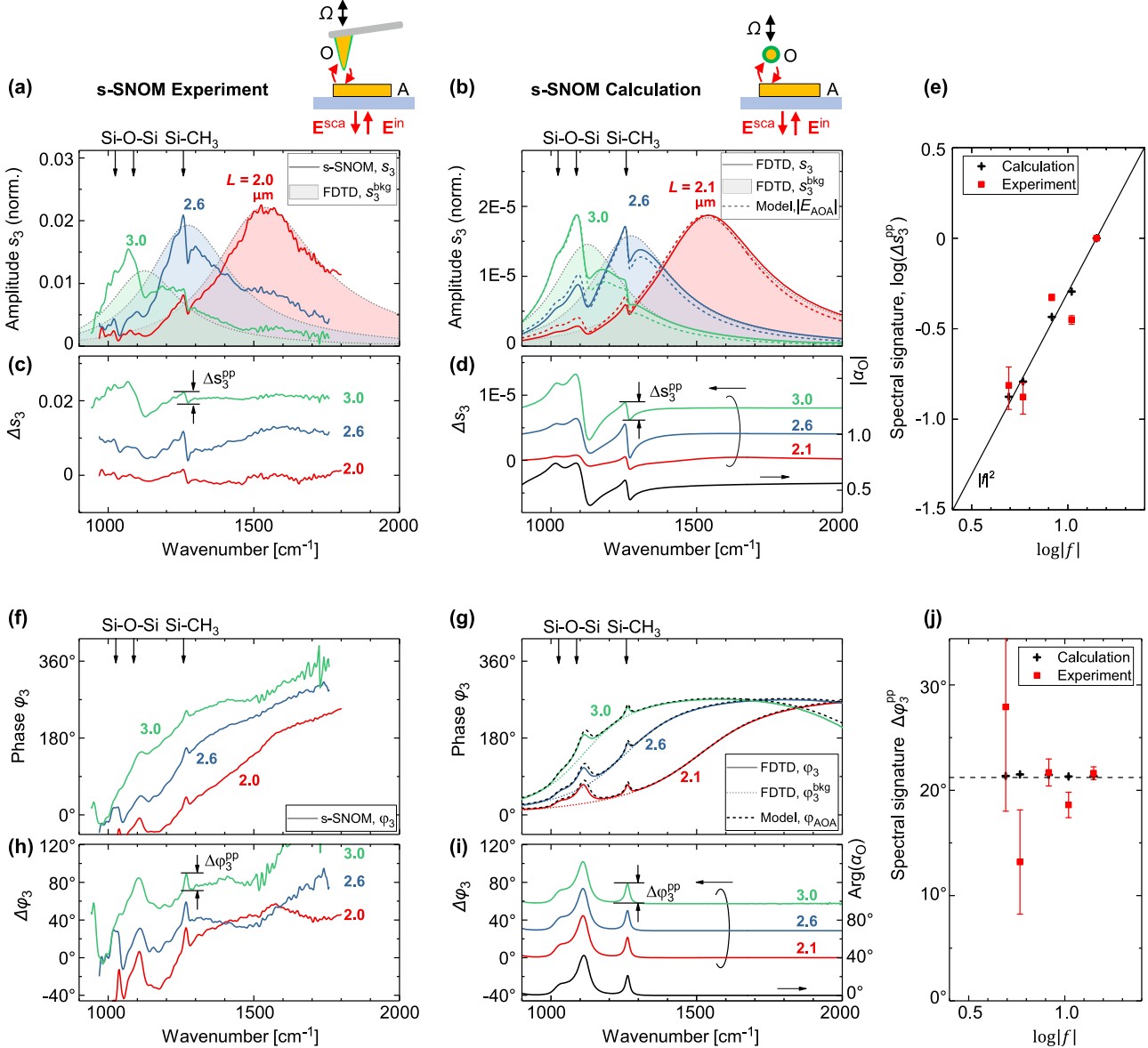

**Fig. 3 | Near-field spectroscopic measurement of field-enhanced molecular scattering. a** Experimental (s-SNOM) amplitude spectra, $s_3(\omega)$, obtained with a PDMS-contaminated AFM tip (O) that interacts with a metal nanorod (A) of different lengths, $L$ (solid lines). Shaded areas show the numerically calculated near-field spectra, $s_3^{bkg}(\omega)$, from (**b**) but scaled to the experimental data (Supplementary Note 1). **b** Numerically calculated (FDTD) amplitude spectra, $s_3(\omega)$, (solid lines), where the AFM tip is described as a core-shell nanoparticle with a 50 nm radius Au core and a 10 nm thick PDMS shell, and aspects of tip vibration and signal demodulation were taken into account (Supplementary Note 1). Shaded areas show the calculated amplitude spectra, $s_3^{bkg}(\omega)$, assuming a non-absorbing dielectric shell. Dashed lines show the magnitude of field-enhanced molecular scattering, $|E_{AOA}(\omega)|$, (Eq. 18, no demodulation considered, scaled to the calculated data). **c**, **d** Isolated spectral signature of the molecular vibrations, $\Delta s_3(\omega) = s_3(\omega) - s_3^{bkg}(\omega)$, from the data in (**a**, **b**) (curves are offset for clarity). The magnitude of the polarizability of the core-shell nanoparticle considered in (**b**), $|\alpha_O(\omega)|$, is shown for reference (black solid line in (**d**)). **e** Parametric representation of the spectral signature of the 1258 cm⁻¹ (Si-CH₃) molecular vibration of PDMS, $\Delta s_3^{pp}$, as obtained from panels

(**a**, **b**), normalized to maximum and plotted against the calculated field enhancement, $|f|$, at 1258 cm⁻¹. **f** Experimental (s-SNOM) and **g** numerically calculated (FDTD) phase spectra, $\varphi_3(\omega)$, (solid lines). The dotted lines in (**g**) show the calculated phase spectra, $\varphi_3^{bkg}(\omega)$, assuming a non-absorbing dielectric shell. Black dashed lines show the phase of field-enhanced molecular scattering, $\varphi_{AOA}(\omega)$ (Eq. 19). **h**, **i** Baseline-corrected phase spectra, $\Delta\varphi_3(\omega) = \varphi_3(\omega) - \varphi_3^{bkg}(\omega)$ from the data in (**f**, **g**) (curves are offset for clarity). The phase of the polarizability of the core-shell nanoparticle, $\text{Arg}\{\alpha_O(\omega)\}$, is shown for reference in (**i**) (black solid line). **j** Parametric representation of the spectral signature of the molecular vibration (Si-CH₃) of PDMS, $\varphi_3^{pp}$, as obtained from panels (**h**, **i**), plotted against the calculated field enhancement, $f$, at 1258 cm⁻¹. Error bars in (**e**, **j**) describe the uncertainty owing to experimental noise (see Methods). Supplementary Fig. 5 shows the spectra for each antenna individually, Supplementary Note 2 details the corresponding spectrally integrated near-field images. The PDMS shell of the core-shell nanoparticle in (**b**) was modeled using tabulated data, and the non-absorbing dielectric shell with constant $\varepsilon = 1.8$. The polarizability of the core-shell nanoparticle was evaluated analytically using Eq. 2 in Supplementary Note 1.

observed in the extinction cross section of SEIRA spectroscopy (Fig. 1). To this end, we performed the same calculation of SEIRA spectroscopy as in Fig. 1b,c, but now considering a CaF₂ substrate, a 60 nm radius PDMS particle and the geometry of the experimental nanorods. We plot the baseline-corrected extinction cross section,

$\Delta\sigma^{ext}(\omega) = \sigma^{ext}(\omega) - \sigma_{bkg}^{ext}(\omega)$, revealing the expected Fano-like vibrational line shapes for different nanorod lengths, $L$ (Fig. 4a). For comparison, we plot the imaginary part of the baseline-corrected experimental near-field spectra, $\Delta\tilde{s}_3(\omega) = \tilde{s}_3(\omega) - \tilde{s}_3^{bkg}(\omega)$, in Fig. 4b (solid lines), where $\tilde{s}_3(\omega)$ is the measured near-field spectrum

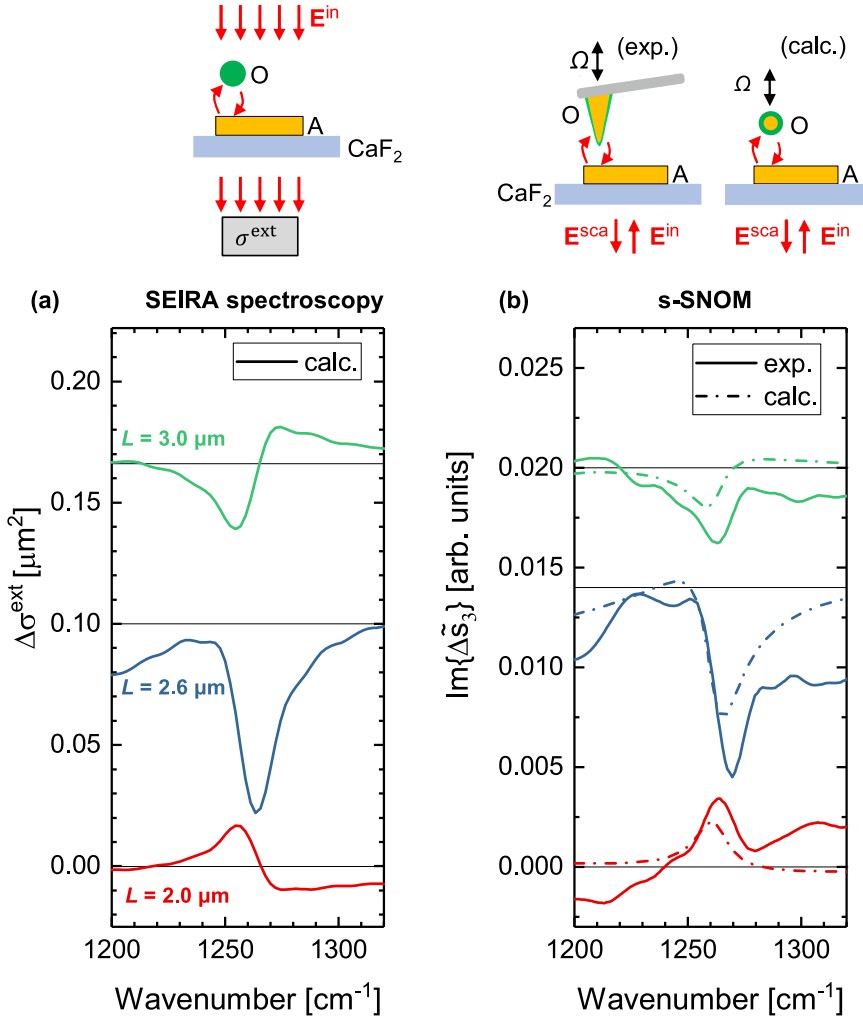

**Fig. 4 | Field-enhanced molecular scattering as measured by s-SNOM yields the vibrational line shapes in the extinction cross section. a** Calculated spectral signature of the molecular vibrations in the extinction cross section of SEIRA spectroscopy, $\Delta\sigma^{ext}(\omega) = \sigma^{ext}(\omega) - \sigma^{ext}_{bkg}(\omega)$, where a Au nanorod antenna of length $L$ couples to a homogeneous PDMS nanoparticle. **b** Imaginary part, $\mathrm{Im}\{\Delta\tilde{s}_3(\omega)\}$, of the baseline-corrected near-field spectra, $\Delta\tilde{s}(\omega) = \tilde{s}_3(\omega) - \tilde{s}^{bkg}_3(\omega)$ (solid lines: using the experimental s-SNOM data from Fig. 3a,f as obtained with a molecule-coated AFM tip, dashed lines: using the calculated s-SNOM data from Fig. 3b, g as obtained with a core-shell nanoparticle with 50 nm Au radius core and 10 nm thick PDMS shell).

obtained with the PDMS-contaminated AFM tip (solid lines in Fig. 3a,f) and $\tilde{s}^{bkg}_3(\omega)$ is the calculated near-field spectrum assuming a core-shell nanoparticle with a non-absorbing dielectric shell (shaded area in Fig. 3a). For further comparison, we also plot the baseline-corrected calculated near-field spectra, $\Delta\tilde{s}_3(\omega) = \tilde{s}_3(\omega) - \tilde{s}^{bkg}_3(\omega)$, in Fig. 4(b) (dashed lines), where $\tilde{s}_3(\omega)$ is the numerically calculated near-field spectrum obtained with the Au-PDMS core-shell nanoparticle (solid lines in Fig. 3(b,g)) and $\tilde{s}^{bkg}_3(\omega)$ is again the calculated near-field spectrum assuming a core-shell nanoparticle with a non-absorbing dielectric shell (shaded area in Fig. 3b). This baseline correction removes the field-enhanced scattering from the dielectric background (i.e., excluding the vibrational response) of the AFM tip. Since the field-enhanced molecular scattering is measured in amplitude and phase in the s-SNOM experiment (Eq. (17)), we can plot the imaginary part of $\Delta\tilde{s}_3$ to produce the spectral signature of the molecular vibrations in analogy to Eq. (9). The so-obtained vibrational line shapes, $\mathrm{Im}\{\Delta\tilde{s}_3(\omega)\}$ (Fig. 4b), are in good agreement with the vibrational line shapes in the extinction cross section of SEIRA spectroscopy, $\Delta\sigma^{ext}$ (Fig. 4a), thus providing experimental evidence that the interference between field-enhanced molecular scattering $\mathbf{E}_{AOA}$ and the incident field can fully explain the vibrational signature in SEIRA spectra.

## Discussion

We have demonstrated theoretically and numerically that the vibrational line shapes in the extinction cross section of SEIRA spectroscopy can be explained by the interference of field-enhanced molecular scattering, $\mathbf{E}_{AOA}$, with the incident field, $\mathbf{E}^{in}$. For an experimental verification of our theory, we employed a nano-FTIR spectroscopy setup operating in transflection mode to measure the field-enhanced molecular scattering, $\mathbf{E}_{AOA}$, at single infrared antennas. We found that the field-enhanced molecular scattering, $\mathbf{E}_{AOA}$, can be measured and scales in intensity with the fourth power of the local field enhancement, $|f|^4$. Projection of the measured nano-FTIR spectra on the imaginary part yielded the vibrational line shapes in SEIRA spectra, providing experimental evidence that the interference between field-enhanced molecular scattering, $\mathbf{E}_{AOA}$, and the incident field, $\mathbf{E}_{in}$, can fully explain the vibrational signature in SEIRA spectra.

Interestingly, in our experiments we observe field-enhanced molecular scattering signatures that are larger than those typically observed in SEIRA spectroscopy. Specifically, with the nanorods shown in Fig. 3, the spectral signature can reach up to 41% of antenna resonance maximum (Fig. 3c). With a gap antenna – affording stronger coupling between molecule and antenna – an

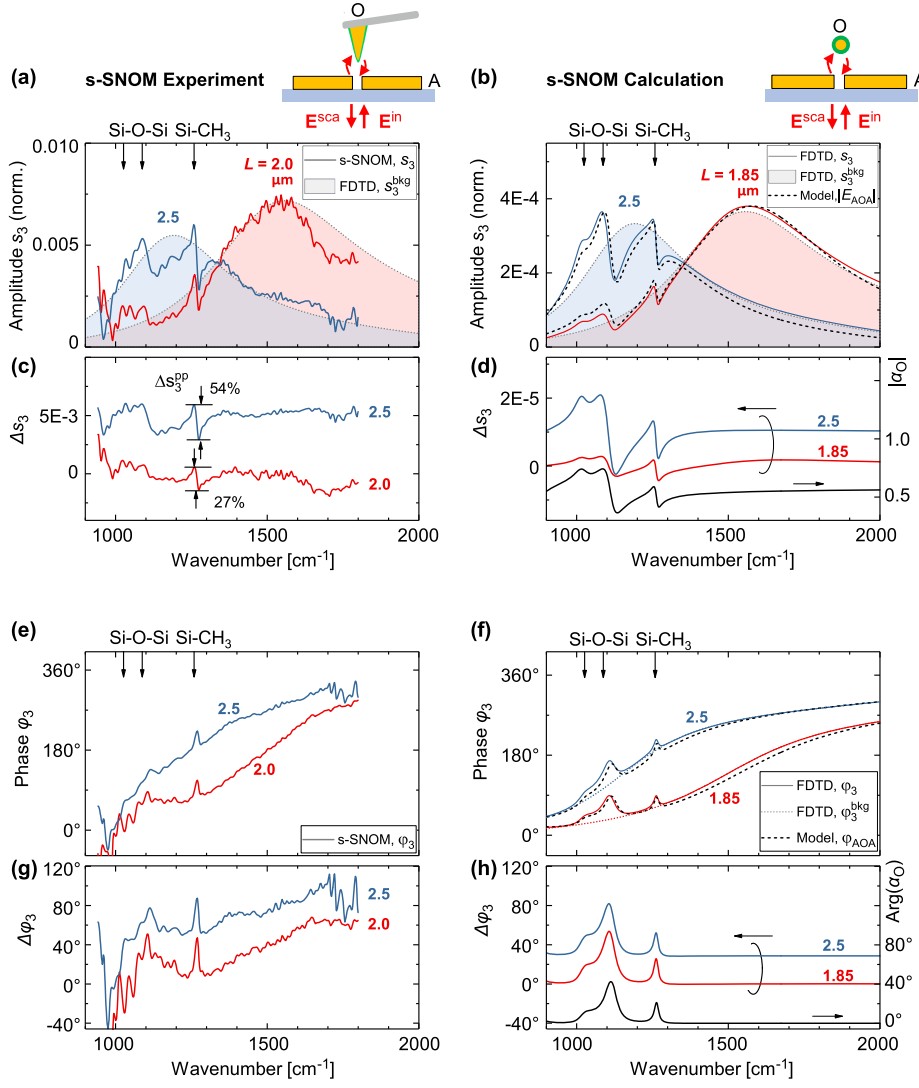

**Fig. 5 | Near-field spectroscopic measurement of field-enhanced molecular scattering with a gap antenna. a** Experimental (s-SNOM) amplitude spectra, $s_3(\omega)$, obtained with a PDMS-contaminated AFM tip (O) that interacts with a gap antenna (A) (pair of single rods identical to those in Fig. 3, of rod length, $L$, and separated by a 100 nm gap) (solid lines). Shaded areas show the numerically calculated near-field spectra, $s_3^{\text{bkg}}(\omega)$, from (**b**), but scaled (Supplementary Note 1). **b** Numerically calculated (FDTD) amplitude spectra, $s_3(\omega)$, (solid lines), where the AFM tip is described as a core-shell nanoparticle as in Fig. 3b. Shaded areas show the calculated amplitude spectra, $s_3^{\text{bkg}}(\omega)$, assuming a non-absorbing dielectric shell. Dashed lines show the magnitude of field-enhanced molecular scattering, $|E_{\text{AOA}}(\omega)|$.

**c**, **d** Isolated spectral signature of the molecular vibrations, $\Delta s_3(\omega) = s_3(\omega) - s_3^{\text{bkg}}(\omega)$, from the data in (**a,b**) (curves are offset for clarity). **d** The magnitude of the polarizability of the core-shell nanoparticle considered in (**b**), $|\alpha_O(\omega)|$, is shown for reference (black solid line). **e** Experimental (s-SNOM) and (**f**) numerically calculated (FDTD) phase spectra, $\varphi_3(\omega)$, (solid lines). **f** The dotted lines show the calculated phase spectra, $\varphi_3^{\text{bkg}}(\omega)$, assuming a non-absorbing dielectric shell. Black dashed lines show the phase of field-enhanced molecular scattering, $\varphi_{\text{AOA}}(\omega)$ (Eq. 19). **g**, **h** Baseline-corrected phase spectra, $\Delta\varphi_3(\omega) = \varphi_3(\omega) - \varphi_3^{\text{bkg}}(\omega)$ (curves offset for clarity) from the data in (**e**, **f**). The phase of the polarizability of the core-shell nanoparticle, $\text{Arg}\{\alpha_O(\omega)\}$, is shown for reference in (**h**) (black solid line).

even larger spectral signature can be observed of up to 54% (Fig. 5c). This high molecular contrast is a result of the demodulation techniques employed in the presented s-SNOM experiment. In the future, the combination of scattered light detection and demodulation techniques could be exploited to develop highly sensitive SEIRA sensors.

Finally, and from a more general perspective, the nano-FTIR spectroscopy setup may be applied in the future for studying coupling phenomena on the nanoscale between individual objects. Particularly, near-field microscopy using functionalized tips could be a useful tool to map SEIRA enhancement in nanophotonic structures.

## Methods
### Elastic light scattering process
We describe the scattered field of the antenna-object system $\mathbf{E}^{\text{sca}}$ as a series of multiple scattering events occurring between the antenna and

the object in the limit of a weakly scattering object (i.e., small object polarizability, $\alpha_O$)[18,19]:

$$\mathbf{E}^{\text{sca}}(\mathbf{r}) = \mathbf{E}_A(\mathbf{r}) + \mathbf{E}_O(\mathbf{r}) + \mathbf{E}_{OA}(\mathbf{r}) + \mathbf{E}_{AO}(\mathbf{r}) + \mathbf{E}_{AOA}(\mathbf{r}) + \cdots . \quad (20)$$

The following expressions may be obtained for the individual terms by considering the tensorial nature of the diagonal antenna polarizability, $\overset{\leftrightarrow}{\boldsymbol{\alpha}}_A$, the object polarizability, $\overset{\leftrightarrow}{\boldsymbol{\alpha}}_O$, and the field enhancement, $\mathbf{f}$ (tensor of components, $f_{ij}$ with $i,j = x, y, z$)[17,19]:

$$\mathbf{E}_A(\mathbf{r}) = k^2 \overset{\leftrightarrow}{\mathbf{G}}(\mathbf{r}, \mathbf{r}_A) \overset{\leftrightarrow}{\boldsymbol{\alpha}}_A \mathbf{E}^{\text{in}}(\mathbf{r}_A), \quad (21)$$

$$\mathbf{E}_O(\mathbf{r}) = k^2 \overset{\leftrightarrow}{\mathbf{G}}(\mathbf{r}, \mathbf{r}_O) \overset{\leftrightarrow}{\boldsymbol{\alpha}}_O \mathbf{E}^{\text{in}}(\mathbf{r}_O), \quad (22)$$

$$\mathbf{E}_{OA}(\mathbf{r}) = k^2 \overleftrightarrow{\mathbf{G}}(\mathbf{r}, \mathbf{r}_O) \overleftrightarrow{\boldsymbol{\alpha}}_O \overleftrightarrow{\mathbf{f}} \mathbf{E}^{in}(\mathbf{r}_A), \tag{23}$$

$$\mathbf{E}_{AO}(\mathbf{r}) = k^2 \overleftrightarrow{\mathbf{G}}(\mathbf{r}, \mathbf{r}_A) \overleftrightarrow{\mathbf{f}}^{T} \overleftrightarrow{\boldsymbol{\alpha}}_O \mathbf{E}^{in}(\mathbf{r}_O), \tag{24}$$

$$\mathbf{E}_{AOA}(\mathbf{r}) = k^2 \overleftrightarrow{\mathbf{G}}(\mathbf{r}, \mathbf{r}_A) \overleftrightarrow{\mathbf{f}}^{T} \overleftrightarrow{\boldsymbol{\alpha}}_O \overleftrightarrow{\mathbf{f}} \mathbf{E}^{in}(\mathbf{r}_A), \tag{25}$$

where $\mathbf{r}_A$ and $\mathbf{r}_O$ are the position of the dipole that represents the antenna and object, respectively, and $\mathbf{r}$ is the point where the electric field is evaluated, $\overleftrightarrow{\mathbf{G}}$ is the Green's tensor in free space, and the symbol T is the transpose of a tensor. Note that the field enhancement tensor is defined as $\overleftrightarrow{\mathbf{f}} = k^2 \overleftrightarrow{\mathbf{G}}(\mathbf{r}_O, \mathbf{r}_A) \overleftrightarrow{\boldsymbol{\alpha}}_A$. Approximate expressions may be obtained by considering the configuration of the numerical calculations. First, the incident field is configured to be $x$-polarized, $\mathbf{E}^{in}(\mathbf{r}_A) = \mathbf{E}_0^{in} \exp(i\mathbf{k} \cdot \mathbf{r}_A) = E_x^{in}(\mathbf{r}_A)\hat{\mathbf{x}}$, and thus parallel to the long axis of the antenna considered in Fig. 1. Second, the electric field at the end and on the long axis of the rod antenna is mainly $x$-polarized (dominance of the one component of the field enhancement factor over the others, $|f_{xx}| \gg |f_{yx}|, |f_{zx}|$). Third, the polarizability of the spherical object is assumed to be isotropic ($\alpha_{O,xx} = \alpha_{O,yy} = \alpha_{O,zz}$), that is, we consider that the object consists of a large number of randomly oriented molecules. Above terms (Eqs. (21)–(25)) may thus be approximated to

$$\mathbf{E}_A(\mathbf{r}) \approx k^2 \mathbf{G}_x(\mathbf{r}, \mathbf{r}_A) \alpha_A E_x^{in}(\mathbf{r}_A), \tag{26}$$

$$\mathbf{E}_O(\mathbf{r}) \approx k^2 \mathbf{G}_x(\mathbf{r}, \mathbf{r}_O) \alpha_O E_x^{in}(\mathbf{r}_O), \tag{27}$$

$$\mathbf{E}_{OA}(\mathbf{r}) \approx k^2 \mathbf{G}_x(\mathbf{r}, \mathbf{r}_O) \alpha_O f E_x^{in}(\mathbf{r}_A), \tag{28}$$

$$\mathbf{E}_{AO}(\mathbf{r}) \approx k^2 \mathbf{G}_x(\mathbf{r}, \mathbf{r}_A) f \alpha_O E_x^{in}(\mathbf{r}_O), \tag{29}$$

$$\mathbf{E}_{AOA}(\mathbf{r}) \approx k^2 \mathbf{G}_x(\mathbf{r}, \mathbf{r}_A) f \alpha_O f E_x^{in}(\mathbf{r}_A), \tag{30}$$

where $\mathbf{G}_x$ is $x$-component of the Green's tensor function, $\mathbf{G}_x = \overleftrightarrow{\mathbf{G}} \cdot \hat{\mathbf{x}}$, and indices $xx$ were omitted for clarity. Considering the much smaller polarizability of the object compared to the antenna, $\alpha_A \gg \alpha_O$, the interaction series in Eq. (20) may be truncated to first order in O. Further assuming large field enhancement, $|f| \gg 1$, it is sufficient to only retain terms $\mathbf{E}_A(\mathbf{r})$ and $\mathbf{E}_{AOA}(\mathbf{r})$ in $\mathbf{E}^{sca}(\mathbf{r})$, where $\mathbf{E}_{AOA}(\mathbf{r})$ describes the field scattered from the object via the antenna after being illuminated by the antenna. Terms $\mathbf{E}_{OA}(\mathbf{r})$ and $\mathbf{E}_{AO}(\mathbf{r})$ – describing a single scattering event between antenna and object – only scale linearly in $f$ and are thus much smaller than the term $\mathbf{E}_{AOA}(\mathbf{r})$ and will be neglected. The scattered field may thus be reduced to

$$\mathbf{E}^{sca}(\mathbf{r}) \approx \mathbf{E}_A(\mathbf{r}) + \mathbf{E}_{AOA}(\mathbf{r}). \tag{31}$$

The scattered field intensity may be similarly approximated. With Eq. (20), $|\mathbf{E}^{sca}|^2$ may be developed as follows (argument $\mathbf{r}$ omitted for brevity):

$$\frac{|\mathbf{E}^{sca}|^2}{|\alpha_O|^2} \approx \underbrace{\mathbf{E}_A^* \cdot \mathbf{E}_A}_{|\alpha_A|^2 \sim 6 \cdot 10^{19}} + \underbrace{\mathbf{E}_A^* \cdot \mathbf{E}_{AOA}}_{|\alpha_A f^2 \alpha_O| \sim 2 \cdot 10^{13}} + \underbrace{\mathbf{E}_A^* \cdot \mathbf{E}_{AO}}_{|\alpha_A f \alpha_O| \sim 4 \cdot 10^{11}}$$

$$+ \underbrace{\mathbf{E}_A^* \cdot \mathbf{E}_{OA}}_{|\alpha_A f \alpha_O| \sim 4 \cdot 10^{11}} + \underbrace{\mathbf{E}_A^* \cdot \mathbf{E}_O}_{|\alpha_A \alpha_O| \sim 8 \cdot 10^9} + c.c., \tag{32}$$

where the series was truncated to first order in O and terms are listed in descending order of importance and $c.c.$ means complex conjugate. By way of example, an estimate for the magnitude is stated below each term in units of $|\alpha_O|^2$ (pure molecular scattering), assuming the values from the numerical calculation shown in the main text (1 nm radius spherical nanoparticle), $2|\alpha_A|/|\alpha_O| \sim 1.6 \cdot 10^{10}$ and $|f| \sim 50$. It is apparent that above series may be reduced by retaining only the first and second term, while incurring an error of only about 4% by omitting terms 3 and 4:

$$|\mathbf{E}^{sca}(\mathbf{r})|^2 \approx \mathbf{E}_A^*(\mathbf{r}) \cdot \mathbf{E}_A(\mathbf{r}) + 2\mathrm{Re}\left\{\mathbf{E}_A^*(\mathbf{r}) \cdot \mathbf{E}_{AOA}(\mathbf{r})\right\}, \tag{33}$$

Note that so far, we have assumed that the object (O) is positioned on the long axis of the nanorod antenna, as it is the case in Fig. 1. If the object (O) is located on top of the nanorod antenna (as it is the case in Fig. 3), then we can make use of the fact that the $z$-component of the field enhancement is dominant above the nanorod ends, $|f_{zx}| \gg |f_{xx}|, |f_{yx}|$. We can then approximate $\mathbf{E}_{AOA}$ by

$$\mathbf{E}_{AOA}(\mathbf{r}) \approx k^2 G_x(\mathbf{r}, \mathbf{r}_A) f_{zx} \alpha_{O,zz} f_{zx} E_x^{in}(\mathbf{r}_A), \tag{34}$$

By substituting $\alpha_O = \alpha_{O,zz}$ (spherical object) and $f = f_{zz}$, we arrive at the same expression for the scattered field, $\mathbf{E}_{AOA}$, as above in Eq. (30).

## Numerical calculation of SEIRA spectra

We provide details on the numerical study presented in Fig. 1 and Supplementary Table 1 and Supplementary Figs. 1–3. We considered a cylindrical gold antenna of 50 nm radius and with round ends. The antenna was placed in vacuum, that is, no substrate was assumed. A spherical nanoparticle (NP) with 10 nm radius was placed on the antenna's long axis with a separation of 20 nm between the antenna end cap and the nanoparticle surface. We used the Lorentzian model to describe the vibrational resonance of the NP material:

$$\varepsilon_O(\omega) = \varepsilon_{bkg} + \varepsilon_{vib} = \varepsilon_{bkg} + \frac{\varepsilon_{Lorentz} \cdot \omega_0^2}{\omega_0^2 - \omega^2 - 2i\gamma\omega}, \tag{35}$$

with background permittivity $\varepsilon_{bkg} = 1.55$, resonance $\omega_0 = 1258\,\mathrm{c\,m^{-1}}$ and linewidth $\gamma = 10\,\mathrm{c\,m^{-1}}$. A Lorentz oscillator strength $\varepsilon_{Lorentz} = 0.015$ was assumed to mimic the Si-CH$_3$ vibrational resonance of PDMS that is probed in the experiments. A larger Lorentz oscillator strength $\varepsilon_{Lorentz} = 0.1$ and a NP radius of 30 nm was assumed in Fig. 1 in the main text to show more clearly the spectral signature of the molecular vibration. We assumed tabulated values for the permittivity of Au[28] for the antenna. We used a commercial software package based on the Finite-difference time-domain (FDTD) method to calculate the relevant quantities of SEIRA spectroscopy (Ansys Lumerical FDTD, Ansys, Inc). For each configuration of the antenna-NP system, we performed a total of four calculations. In all calculations, plane-wave illumination of the antenna with the polarization along the antenna axis was assumed, and the cross sections were calculated by considering the outward flowing net power. A mesh of 5 nm was assumed around the antenna and the mesh was refined at the NP to 1 nm (NP radius $\geq 10$ nm) and 0.5 nm (NP radius < 10 nm). First, we calculated the extinction and scattering cross section, $\sigma^{ext}$ and $\sigma^{sca}$, of the antenna-NP system by considering an absorbing NP (Eq. (35)). Second, we calculated the extinction and scattering cross section, $\sigma_{bkg}^{ext}$ and $\sigma_{bkg}^{sca}$, of the antenna-NP system by considering a non-absorbing NP ($\varepsilon_{Lorentz} = 0$ in Eq. (35)). From these two calculations we obtained the spectral signature of the molecular vibration in extinction, $\Delta\sigma^{ext} = \sigma^{ext} - \sigma_{bkg}^{ext}$, and scattering cross section, $\Delta\sigma^{sca} = \sigma^{sca} - \sigma_{bkg}^{sca}$. Third, we calculated the 3D near-field distribution of the antenna only (i.e., the NP was removed). From this recorded near-field distribution we obtained the dipole moment of the antenna, $p_A$, following the approach from ref. 10. Fourth, we determined the

incident field at the center of the antenna, $E_x^{in}$. We then calculated the antenna polarizability, $\alpha_A = p_A / E_x^{in}$. From the recorded near-field distribution (3rd simulation), we further obtained the $x$-component of the near fields produced by the antenna at the center of the NP, $E_{NF,x}$, from which we calculated the corresponding field enhancement factor, $f = E_x / E_x^{in}$ (the $y$ and $z$-components can be neglected, see section above). Note that for NP radius of 10 nm or smaller, it is sufficient to evaluate the field enhancement at the center of the NP. Only in case of NP radius of 30 nm, where the inhomogeneity of the antenna near fields become significant, the average field enhancement factor $f_{avg}^2 = \frac{1}{V_{np}} \int_{V_{np}} f^2 dV$ was calculated[15]. Finally, we evaluated the object polarizability analytically according to $\alpha_O = 4\pi a^3 \frac{\varepsilon_O - 1}{\varepsilon_O + 2}$, where $a$ is the radius and $\varepsilon_O$ is the permittivity of the object. We then straightforwardly calculated the spectral signature of the molecular vibration from the scattering model, $\sigma_{vib}^{ext}$ and $\sigma_{vib}^{sca}$, by inserting above quantities in Eq. (13) in the main text.

We note that small antenna geometries such as the rod antenna considered here are accurately described by their dipole moment (to less than 1% error in the scattering cross section). This approximation may prove to be insufficient for a quantitative prediction of SEIRA spectroscopy in case of larger antenna structures, requiring an adaptation of the scattering model. A possible approach is outlined in ref. 15. Nevertheless, it was shown that the picture of the point polarizability may still provide a qualitative description of the scattering of large antenna geometries (such as in near-field microscopy[18,29]), possibly extending the applicability of the scattering model to qualitatively describe SEIRA spectroscopy in this regard. We further note that the presented scattering model applies for both thick (scattering dominated) and thin (absorption dominated) rod antennas because knowledge of the forward scattered field alone is sufficient to evaluate the extinction cross section with the optical theorem, and such variation of the antenna polarizability $\alpha_A$ will be explored in the future.

## Sample fabrication

We fabricated Au rectangular-shaped nanorods of different lengths on a CaF$_2$ substrate via high-resolution electron-beam lithography, where the nanorod length was chosen in the range of 1.5 to 3.5 μm in steps of 0.5 μm to yield a fundamental dipolar resonance in the mid-IR spectral range. Polymethyl methacrylate (PMMA) was spin-coated onto the substrate at 4000 rpm as the electron-sensitive polymer. The PMMA was subsequently covered by a 2 nm thick layer of Au for enabling lithography on the insulating substrate. After the electron-beam assisted writing of the nanorods, Au was chemically etched (5 s immersion in KI/I2 solution) and the PMMA was developed in methyl isobutyl ketone: isopropanol 1:3. Finally 5 nm layer of Ti was deposited by electron beam evaporation followed by thermal evaporation of 50 nm of Au. The lift-off of the nanorods was done by immersing the sample in acetone overnight. AFM characterization showed that the Au nanorods measured 60 nm in height, 250 nm in width, and the measured nanorod length is indicated in Fig. 3a and Supplementary Fig. 5.

## Near-field experiment

We modified a commercial s-SNOM system (NeaSCOPE, attocube systems AG) to allow for sample illumination and light collection from below the sample (transflection-mode s-SNOM). This modality was recently demonstrated for near-field mapping of IR-resonant antennas (in liquid)[24]. The transflection-mode geometry allows for efficient excitation of the antenna structures on the sample. At the same time, direct excitation of the AFM tip is largely avoided (because the long axis of the AFM tip is in $z$-direction (vertical) and thus oriented in the direction of light propagation), allowing for the use of metallic AFM tips as strong scatterer of local near fields. Here, we introduce and describe spectroscopic transflection-mode measurements of single IR-resonant antennas based on nano-FTIR. In detail, we used standard Pt-Ir coated (metallic) AFM tips (NCPt arrow tip, Nanoworld) to map the

rod antennas. These AFM tips are known to be contaminated by a thin layer of PDMS (Polydimethylsiloxane)[23,24], which we use here to mimic the molecules in a traditional SEIRA spectroscopy experiment. A parabolic mirror (focal length 8 mm, NA 0.44) was used to focus the broadband infrared beam of the nano-FTIR laser at normal incidence through a CaF$_2$ window (sample holder) and the substrate. The polarization of the illuminating beam was chosen to be parallel to the nanorod axis for efficient excitation of the dipolar resonance. The antenna scattered light was collected from below the sample with the same parabolic mirror, analyzed with the interferometer of the nano-FTIR module and detected with an MCT detector (InfraRed Associates, Inc.). Specifically, the antenna scattered field $\mathbf{E}^{sca}$ was superposed with the external reference field of the nano-FTIR interferometer, $\mathbf{E}^{ref}$, yielding a detector signal proportional to $I_d \propto |\mathbf{E}^{sca}|^2 + |\mathbf{E}^{ref}|^2 + 2\text{Re}\left\{ \mathbf{E}^{ref*} \cdot \mathbf{E}^{sca} \right\}$. With Eqs. (3),(4), the detector signal can be approximated as

$$I_d \propto |\mathbf{E}_A|^2 + 2\text{Re}\left\{ \mathbf{E}_A^* \cdot \mathbf{E}_{AOA} \right\} + 2\text{Re}\left\{ \mathbf{E}^{ref*} \cdot \mathbf{E}_A \right\} + 2\text{Re}\left\{ \mathbf{E}^{ref*} \cdot \mathbf{E}_{AOA} \right\} + |\mathbf{E}^{ref}|^2.$$
$$(36)$$

To suppress the direct antenna scattering, $\mathbf{E}_A$, in Eq. (36), the AFM tip was vertically vibrated sinusoidally with amplitude $\Delta d \sim 100$ nm and frequency $\Omega = 256$ kHz, $d = d_0 + \Delta d \cos \Omega t$. Demodulation of the detector signal at a frequency $n\Omega$ ($n = 3$) already suppresses terms $|\mathbf{E}_A|^2$, $2\text{Re}\left\{ \mathbf{E}^{ref*} \cdot \mathbf{E}_A \right\}$ and $|\mathbf{E}^{ref}|^2$ in Eq. (36). However, signal demodulation alone cannot suppress the interference of field-enhanced molecular scattering, $\mathbf{E}_{AOA}$, with the direct antenna scattering, $\mathbf{E}_A$ (term $2\text{Re}\left\{ \mathbf{E}_A^* \cdot \mathbf{E}_{AOA} \right\}$ in Eq. (36)). To suppress this term and isolate $2\text{Re}\left\{ \mathbf{E}^{ref*} \cdot \mathbf{E}_{AOA} \right\}$, it is further necessary to modulate the phase of $\mathbf{E}^{ref}$ by translating the reference arm mirror[30]. Note that translation of the reference arm mirror also provides for spectral analysis of $\mathbf{E}_{AOA}$ following the Fourier-transform approach: (i) Recording of the demodulated detector signal as a function of the reference mirror position to yield an interferogram, (ii) Removal of the global phase offset to determine the sign of the interferogram signal, (iii) Application of Tukey window with parameter $\alpha = 0.1$, and (iv) Zero-filling by a factor of 3 (v) Performing a fast Fourier transform to obtain the raw near-field amplitude and phase spectra, $s_3^{raw}(\omega)$ and $\varphi_3^{raw}(\omega)$.

The raw near-field spectra of antennas were normalized to a spectrally flat reference. To this end, we fabricated a large (100 μm x 100 μm) metal (Au) patch that was located adjacent to the antennas on the same sample. We then carried out the following steps. First, the AFM tip was moved to the center of the metal patch and approached to the patch surface. By doing so, the incident beam was focused on, and reflected at, the metal patch. Note that the AFM tip was shielded by the large metal patch and thus did not play a role. We then recorded the DC signal of the MCT detector (using the DC output at the detector amplifier and selecting the signal O0 in the microscope software) and acquired an interferogram. Applying the same steps as above yielded a DC amplitude and phase reference spectrum, $s_{DC}^{ref}(\omega)$ and $\varphi_{DC}^{ref}(\omega)$. The normalized near-field spectra, $\tilde{s}_3(\omega) = s_3(\omega) e^{i\varphi_3(\omega)}$, were then obtained with

$$s_3(\omega) = s_3^{raw}(\omega) / s_{DC}^{ref}(\omega) \text{ and } \varphi_3(\omega) = \varphi_3^{raw}(\omega) - \varphi_{DC}^{ref}(\omega) + 180°. \quad (37)$$

Normalization of the amplitude spectra removed the nano-FTIR source spectrum. Normalization of the phase spectra provided an absolute measurement of the scattering phase of the antenna that corrects for the phase shifts accumulated by the illuminating and scattered beam when traversing the substrate, thus allowing for direct comparison of the phase spectra among different antennas. Note that in the reference measurement, the illuminating beam is subjected to a

phase shift of 180° when reflecting from the metal patch, thus we explicitly added this phase shift to $\varphi_3(\omega)$ in Eq. (37). Because of this normalization procedure, the phase relation between the measured spectra, $\widetilde{s}_3(\omega)$, and the incident field, $\mathbf{E}_{in}$, is fixed to zero, and thus the vibrational line shapes can be directly obtained by simply plotting the imaginary of $\widetilde{s}_3(\omega)$, as done in Fig. 4b. To obtain near-field spectra over a large spectral range, we performed above experiment for different settings of the nano-FTIR laser (labeled settings B, C and D in the software, providing coverage from 850 to 1450 cm$^{-1}$, 1200 to 1900 cm$^{-1}$ and 1500 to 2200 cm$^{-1}$, respectively). We stitched together the individual spectra to a full spectrum by employing a smooth transition at wavenumber 1350 cm$^{-1}$ (between B and C) and 1640 cm$^{-1}$ (between C and D), following the protocol from ref. 31.

The error bars in Fig. 3e,j were obtained as follows. We estimated the measurement noise by calculating the root-mean-square of the complex-valued signal in the near-field spectra, $ns = \mathrm{RMS}\left[s_3^{\mathrm{raw}}(\omega)\exp i\varphi_3^{\mathrm{raw}}(\omega)\right]$, in a range of 2600 and 3500 cm$^{-1}$, where no near-field signal is expected as it is outside the spectral range of the nano-FTIR source. It can be shown that the error in the normalized amplitude and phase signal can be obtained by $\mathrm{err}\left[s_3(\omega)\right] = ns/\left(\sqrt{2}s_{\mathrm{DC}}^{\mathrm{ref}}(\omega)\right)$ and $\mathrm{err}\left[\varphi_3(\omega)\right] = ns/s_3^{\mathrm{raw}}(\omega)$, respectively.

## Data availability
Data underlying the results presented in this paper are not publicly available at this time but may be obtained from the authors upon request. Source data for Figs. 1, 3–5 and Supplementary Figs. 1–3. All raw nano-FTIR data generated during the presented study are available at https://doi.org/10.5281/zenodo.13145195.

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

## Acknowledgements
The authors thank Irene Dolado Lopez for providing the antenna sample and Javier Aizpurua, Ilia Rasskazov and Scott P. Carney for fruitful discussions. This work was financially supported by: Grant CEX2020-001038-M funded by MICIU/AEI /10.13039/501100011033. Grant PID2020-115221GA-C44 (SNOMCELL) funded by MICIU/AEI /10.13039/501100011033 (M.S.). Grant PID2021-123949GB-I00 (NANOSPEC) funded by MICIU/AEI /10.13039/501100011033 and by ERDF/EU (R.H.).

## Author contributions

D.V. performed the measurements. M.S. performed the numerical calculations. D.V., C.M.-E. and M.S. performed data analysis. C.M.-E., R.H. and M.S. developed the theory. R.H. and M.S. wrote the manuscript with input from all authors. All authors contributed to scientific discussions.

## Competing interests

R.H. is a co-founder of Neaspec GmbH, which now is a part of Attocube AG, a company producing s-SNOM systems, such as the one used in this study. The remaining authors declare no competing interests.
