## [Peer Review File · Nature Communications]

Experimental verification of field-enhanced molecular vibrational scattering at single infrared antennasREVIEWERS' COMMENTS:

Reviewer #1 (Remarks to the Author):

The authors have demonstrated the importance of the scattering components in the surface-enhanced infrared absorption (SEIRA) spectra. The origin of the ultrahigh sensitivity and the interpretation of the complex line shapes are the two main challenges in SEIRA. The consider the model interaction between a molecular vibration and a metal antenna in the presence of incident light in the infrared. Please consider following comments and suggestions.

(1) Assuming the molecular resonances and the antenna resonance are overlapped, how the field-enhanced molecular scattering alone describe the line shapes in a SEIRA spectrum? I suggest the authors discuss different cases of the line shapes in the main text.

(2) If the antenna is not scattering-dominated, does the conclusion that molecular scattering alone describe the magnitude or line shapes of the SEIRA spectra?

(3) The authors prepared a PDMS-coated metallic tip as a model system to check the validity of the molecular scattering alone describe the SEIRA spectra. However, the coupling between the metal tip and the metal rod is strong. As a consequence, the near-field scattering signals from the s-SNOM configuration results two originals, one is from the molecules, the others are from the near-field coupling from the metal tip-metal antenna. The authors could discuss the influence from the latter to the former.

Reviewer #2 (Remarks to the Author):

The manuscript investigates the contribution of molecular scattering to the scattering amplitude in scattering nearfield infrared spectroscopy. The investigation is done with an analytical model - assuming an isotropic polarizability of a molecular point-dipole and considering multiple interferences between a resonant plasmonic nanorod and the molecular object. Theory and experiments are done for the backscattering geometry with illumination from the substrate backside. It turns out that the molecular scattering cross section appears as enhanced by the square (!) of the nearfield enhancement f in the scattering amplitude spectrum, which means that a measured intensity signal would be enhanced by the 4th power of f , like in SERS. FDTD simulation corroborate the results.

This is an interesting result. Its importance would be strengthened if some points become better explained:

Because the geometry of the set up is very different to what the authors call "typical" SEIRA, the "typical" SEIRA set up (where extinction in a simple forward scattering usually is detected) should be explained. Furthermore, depending on the gaps between the objects, various interferences might influence the molecular SEIRA signal. In that sense, the manuscript shows only one example, however a relevant one for the nearfield scattering technique. To what extent the result is valid also for "typical" SEIRA?

The experimental verification (Fig. 3c and 3f) suffers a little bit from the lack of error bars. In Fig. 3f, the experimental points seem to contradict the theory.

I recommend a minor revision and publication of the interesting paper.

Reviewer #3 (Remarks to the Author):

Recommendation: This paper is not recommended because the physics is incorrect, and the research lacks a control experiment.

Comments:

The experiment described is the detection of the absorption of the PDMS layer on an NSOM tip using a gold nanoparticle antenna to amplify the signal. Broad claims of a factor of 10^{13} enhancement are not supported in the manuscript. Control experiments are needed to verify several claims.

Experimental deficiency: Line 292 "Both near-field probe and antenna were illuminated from below in a transflection geometry with the weakly focused beam from a mid-infrared broadband laser (see Methods). Illumination from below ensured efficient excitation of the antenna while avoiding direct excitation of the near-field probe." What is the control experiment to ensure that there is no measurement of the PDMA directly from the probe? How do we know that the detected light is only the scattered light from the nanoparticle antenna?

In addition, the theory lacks rigor. The authors have failed to distinguish the enhancement

factors that arose in two steps: the local field enhancement factor, which is the enhancement of the incident field broadcast by the antenna, and the object scattering enhancement factor, which is the enhancement of the infrared signal by the antenna. There are a few reasons to separate the two enhancement factors.

1. Because the scattered field from the antenna is evanescent from the surface by localized surface plasmon, the enhancement factor f is a function of distance, which is not included in the theory of this paper. The first enhancement factor is the local field enhancement factor, and the local field is detected by the molecule, it is in near field; the second enhancement factor is the enhanced molecular signal from the antenna to the detector, which is in the far field. As a result, the two enhancement factors should not be the same. This also means that their maximum wavelength of enhancement is different significantly reducing any possible synergy.

2. The increase in local field can result in an increase in absorbance up to a saturation limit. Unlike the case of Raman scattering, where there can be many orders of magnitude of dynamic range in the enhancement, even absorption of 0.01 can only be increased by a circa two orders of magnitude before saturation occurs.

3. Furthermore, the incident field is a plane wave, as defined in Equation 1. However, the scattered wave from the object is a spherical wave for an oscillating dipole, and it is from a much closer source for the antenna compared to the incident light. In this case, the geometric factor is different, the two enhancement factors may also be different in phase, which means, at one frequency, the first f reaches the maximum, the second one may be negligible. The molecular absorption frequency must have a relationship to the surface plasmon frequency in order for a resonance to be observed. This is not discussed.

Therefore, in Equation 6, there should be a function of the observer location r in the enhancement factor, and maybe only one factor of f contributes to the field enhancement, leading to f^2 as the maximal enhancement in absorption intensity, not f^4 . Moreover, the phase of the antenna particle interaction may also be different further complicating the possible observations. There is no physical justification for the f^4 enhancement.

Minor problems:

Line 68, please explain “the quantity” mentioned in “Current SEIRA models do not directly connect to the quantity that is measured in SEIRA”. The sentence is not clear.

Line 131, the authors have mentioned the x direction. It is better to show the coordinate on Figure 1a to define the coordinate system.

Line 144, citation format for ref. 14 and 16 is different from others.

Line 245, in equations 17 and 18, is the background subtracted?

Line 345, are there any experimental results for the control experiment?

Line 372, should be 1.8 ± 0.4 rather than 1.80 ± 0.40 , should use the error to round the sig figs.

Response to Reviewer 1:

We thank the reviewer for the positive evaluation and comments to improve the clarity of our manuscript.

1. Assuming the molecular resonances and the antenna resonance are overlapped, how the field-enhanced molecular scattering alone describe the line shapes in a SEIRA spectrum? I suggest the authors discuss different cases of the line shapes in the main text.

The field enhanced molecular scattering alone can describe the line shapes in a SEIRA spectrum that are obtained for various tuning configurations of the antenna. We had already briefly analyzed different off-resonant cases where the length of the rod antenna was tuned, which is displayed in Extended Data Fig. 2. To address the reviewer's question, we now give these results more prominence and discuss the various line shapes in the main text. However, since the manuscript is already rather long, we prefer to keep this discussion brief.

In the revised manuscript we add the following discussion on page 11

Briefly, the spectral signature in the extinction (scattering) cross section resembles a peak (dispersive) line shape for short rod antennas. In this case, the antenna is driven far below resonance, the field enhancement is nearly real-valued ($\text{Arg}(f) \sim 0$)¹⁵ and Eq. (14) can be approximated by $\sigma_{\text{vib}}^{\text{ext}} = k|f|^2 \text{Im}\{\alpha_{\text{O}}\}$, $\sigma_{\text{vib}}^{\text{sca}} = \frac{k^4}{3\pi} |f|^2 |\alpha_{\text{A}}| \text{Re}\{\alpha_{\text{O}}\}$. As the rod antenna length is increased, the phase of the field enhancement $\text{Arg}(f)$ increases and a transition to a dip is obtained (antenna resonance tuned to molecular resonance, $\text{Arg}(f) = \pi/2$) and further towards a peak (inverse dispersive) line shape is obtained. In the latter case, the antenna is driven far above resonance, $\text{Arg}(f) \rightarrow \pi$ and Eq. (14) can be approximated by $\sigma_{\text{vib}}^{\text{ext}} = k|f|^2 \text{Im}\{\alpha_{\text{O}}\}$, $\sigma_{\text{vib}}^{\text{sca}} = -\frac{k^4}{3\pi} |f|^2 |\alpha_{\text{A}}| \text{Re}\{\alpha_{\text{O}}\}$.

2. If the antenna is not scattering-dominated, does the conclusion that molecular scattering alone describe the magnitude or line shapes of the SEIRA spectra?

This is an excellent question. The answer is yes. The optical theorem relates the extinction of the antenna-object system to the forward scattered field alone. It is not required that antenna scattering dominates antenna extinction.

Preliminary results show that our model delivers good prediction even for antennas of small diameter where antenna absorption dominates over antenna scattering. Please see Figure R1 below showing an excerpt of these preliminary results. The dashed lines (our model from Eq. 14) fit very well to the numerically calculated cross sections (solid lines) for a range of antenna diameters. In a future regular paper, we will explore this aspect in detail, but we feel that such a study would go beyond the scope of the presented manuscript.

Figure R1: Spectral signature of the molecular vibration in the extinction, scattering and absorption cross section of SEIRA.

We added a sentence to the revised manuscript

We further note that the presented scattering model applies for both thick (scattering dominated) and thin (absorption dominated) rod antennas because knowledge of the forward scattered field alone is sufficient to evaluate the extinction cross section with the optical theorem, and such variation of the antenna polarizability α_A will be explored in the future.

3. The authors prepared a PDMS-coated metallic tip as a model system to check the validity of the molecular scattering alone describe the SEIRA spectra. However, the coupling between the metal tip and the metal rod is strong. As a consequence, the near-field scattering signals from the s-SNOM configuration results two originals, one is from the molecules, the others are from the near-field coupling from the metal tip-metal antenna. The authors could discuss the influence from the latter to the former.

The reviewer is correct. The coupling of the antenna to the metal core of the tip is real, however, it turns out that it is not as large as assumed to be. This can be seen with the results of our numerical study in Fig. 3(b). Specifically, our numerical model describes the tip in the experiment -- a micrometer sized conical structure in the experiment -- by a metallic nanoparticle that is coated with a molecular layer. Remarkably, this description -- as crude as it may be -- already reproduces well the experimental spectra (Fig. 3(a)). To support our argument, here we show a numerical simulation where the tip is modeled as a 10 μm long cone that is covered by a thin molecular layer, which is a more accurate description of the real tip geometry in the experiment. The results (Figure R2 below) show that the spectral signature of the molecular vibration is clearly seen with the conical tip and of similar magnitude compared to a nanoparticle tip. We conclude that that the metal tip-metal antenna coupling is comparable to the molecule-metal-antenna coupling and that this coupling is taken into account.

Figure R2: Numerically calculated amplitude spectra of a 2.5 μm long rod antenna using a nanoparticle metallic tip (red line) as considered in the main text, and a large conical metallic tip (blue line, 10 μm long and with a circular base of 2.8 μm radius).

We added the following explanation to the Methods section on page 23 of the revised manuscript.

We note that both the metal core and molecular layer contribute to the scattered light, E_{AOA} , in that scattering by the metal core via the antenna after being illuminated by the antenna essentially probes the field enhancement of the antenna²⁰, which is the reason why the antenna resonance is seen so clearly in Fig. 3(b). Conversely, scattering via the molecular layer via the antenna after being illuminated by the antenna probes the molecular vibration. Both scattering contributions add up to yield the spectra shown in Fig. 3(b) and are modelled here by introducing an effective object polarizability, α_0 , as defined above in Eq. (38). From the good agreement with the experimental data in Fig. 3(a), we conclude that in our specific experiment the molecule-coated tip is well described by a core-shell nanoparticle. Specifically, scattering of the molecular layer is significant compared to the scattering of the metal core of the tip, and the weight of the latter is no more than the scattering of a small metal nanoparticle with the size of the tip apex.

Response to Reviewer 2:

We thank the reviewer for the positive evaluation and input to improve our manuscript.

The manuscript investigates the contribution of molecular scattering to the scattering amplitude in scattering nearfield infrared spectroscopy. The investigation is done with an analytical model - assuming an isotropic polarizability of a molecular point-dipole and considering multiple interferences between a resonant plasmonic nanorod and the molecular object. Theory and experiments are done for the backscattering geometry with illumination from the substrate backside. It turns out that the molecular scattering cross section appears as enhanced by the square (!) of the nearfield enhancement f in the scattering amplitude spectrum, which means that a measured intensity signal would be enhanced by the 4th power of f , like in SERS. FDTD simulation corroborate the results. This is an interesting result. Its importance would be strengthened if some points become better explained:

We thank the reviewer for recognizing the importance of our findings. Indeed, the quadratic scaling of the scattering amplitude with the field enhancement is a key result of our work, that we hope can lead to deeper insights into the mechanism of SEIRA and improved performance of SEIRA sensors.

1. Because the geometry of the set up is very different to what the authors call "typical" SEIRA, the "typical" SEIRA set up (where extinction in a simple forward scattering usually is detected) should be explained.

We follow the reviewer suggestions and introduce the following clarifications in the manuscript text.

On page 3 of the revised manuscript, we added

In this typical configuration, light extinction is detected in the direction of the incident beam, which describes a spectroscopy experiment done in transmission with a transparent substrate.

On page 4, we now write

In Fig. 1(a), we describe the typical SEIRA configuration, where extinction is detected in direction of the incident field, in form of a scattering process [...]

2. Furthermore, depending on the gaps between the objects, various interferences might influence the molecular SEIRA signal. In that sense, the manuscript shows only one example, however a relevant one for the nearfield scattering technique. To what extend the result is valid also for "typical" SEIRA?.

In our manuscript, we focus on the case where one seeks to detect the molecular resonance of an individual small particle. This task is one of the typical applications of a SEIRA experiment, which our model can describe very well. We briefly comment on the extension of our model to other SEIRA configurations.

First, the size of the antenna. We note that both the antenna and object are approximated as point polarizabilities. In case of small antennas such as $(\lambda/2)$ rod antennas, this is a good approximation because the dipole moment alone of a rod antenna describes the antenna scattering to an error of less than 1%. cursory exploration of other antenna geometries shows that this error may become significant as the size of the antenna becomes large compared to the wavelength. For example, in case of a gap antenna (measuring roughly the wavelength λ in length), this error is of the order of 10%. This aspect will be explored in a future regular paper. That being said, depending on the application, even very large antenna geometries can still be qualitatively well described by a point polarizability. This is for example

the case in near-field microscopy where the large conically-shaped tip of an atomic force microscopy has the function of the antenna and it can still be described as a point polarizability despite its large size at infrared frequencies. See for example reference Sun, J., Carney, P. S. & Schotland, J. C. Strong tip effects in near-field scanning optical tomography. *Journal of Applied Physics* 102, 103103 (2007).

Second, SEIRA also often aims at detecting thin molecular films. We anticipate that our model could be adapted to this case by introducing suitable integration over the object susceptibility. Rezus et al. have outlined a strategy to achieving this goal, see Rezus, Y. L. A. & Selig, O. Impact of local-field effects on the plasmonic enhancement of vibrational signals by infrared nanoantennas. *Opt. Express* 24, 12202 (2016).

Third, the relevance of higher-order multiple scattering. When the distance between antenna and object is small, there is a possibility for higher-order multiple scattering events between the antenna and the object. We note that the object polarizability appears to second order (or higher) in higher order terms such as AOAOA. Since our model is assumes a weak object polarizability, e.g. $\alpha_0 \sim 10^{-24} \text{ m}^3$ for a 1 nm radius particle, these terms are very small compared to terms linear in α_0 and can thus be neglected even in the case that field enhancement $f \sim 100$ is large (e.g. close to the antenna). Therefore, the scattering between antenna and object should be well described as a double scattering process, AOA.

To clarify on the aspects above, we now write on page 5 of the revised manuscript

Note that terms to second or higher order in α_0 can be neglected owing to the assumption of small object polarizability, α_0 .

To clarify on the aspects above, we now write in the Methods on page 21 of the revised manuscript

We note that small antenna geometries such as the rod antenna considered here are accurately described by their dipole moment (to less than 1% error in scattering). This approximation may prove to be insufficient for a quantitative prediction of SEIRA in case of larger antenna structures, requiring an adaptation of our model. A possible approach is outlined in ref. 15. Nevertheless, it was shown that the picture of the point polarizability may still provide a qualitative description of the scattering of large antenna geometries (such as in near-field microscopy^{18,24}), possibly extending the applicability of our model to qualitatively describe SEIRA in this regard.

3. The experimental verification (Fig. 3c and 3f) suffers a little bit from the lack of error bars. In Fig. 3f, the experimental points seem to contradict the theory.

I recommend a minor revision and publication of the interesting paper..

The reviewer makes a great suggestion. We have added error bars to the plots in Fig. 3c and 3f. It is now clearly seen that the data points that deviate most from the expected value (Fig. 3f) have actually a very large experimental error and thus should not be given much importance. We nevertheless included them, mainly to show more data points in the region of small field enhancement in Fig. 3c. In a nutshell, the experimental errors is due to the limited SNR of nano-FTIR, which becomes noticeable when the field enhancement of the antennas is small. Therefore, the error bars show that theory is not contradicted, rather they show that the measurement accuracy is limited for the two left most data points.

In the revised manuscript, we also added the following text to the Methods section

The error bars in Fig. 3(c,f) were obtained as follows. We estimated the measurement noise by calculating the root-mean-square of the complex-valued signal in the near-field spectra, $ns = \text{RMS}[s_3^{\text{raw}}(\omega) \exp i\varphi_3^{\text{raw}}(\omega)]$, in a range of 2,600 and 3,500 cm^{-1} , where no near-field signal is expected as it is outside the spectral range of the nano-FTIR source. It can be shown that the error in the normalized amplitude and phase signal is obtained by $\text{err}[s_3(\omega)] = ns / (\sqrt{2} s_{\text{DC}}^{\text{ref}}(\omega))$ and $\text{err}[\varphi_3(\omega)] = ns / s_3^{\text{raw}}(\omega)$, respectively.

Response to Reviewer 3:

We thank the reviewer for the time to evaluate our manuscript. However, we strongly disagree with the reviewer's comments because these comments are in direct contradiction with a vast body of established literature, where the field enhancement effect is described. Specifically, Reviewer #3 makes the following general claims that are all refuted by literature (see table). To be precise and clear on this aspect, we provide a detailed point-by-point response after the table.

Reviewer #3	Literature
“There is no physical justification for the f^4 enhancement.”	Incredibly false statement and refuted by established literature including textbooks. “[...] we find that the Raman-scattered intensity scales linearly with the excitation intensity I_0 and that it depends on the factor” $\left [1 + f_2(\omega_R)][1 + f_1(\omega)] \right ^2.$ “Provided that $\omega_R \pm \omega$ is smaller than the spectral response of the metal nanostructure, the Raman-scattering enhancement scales roughly with the fourth power of the electric field enhancement.” “Notice that the theory outlined [above] is not specific to Raman scattering but applies also to any other linear interaction, such as Rayleigh scattering and fluorescence.” -- Rayleigh scattering is exactly the process that we study in our manuscript. [Novotny, L. & Hecht, B. Principles of nano-optics. (Cambridge University Press, 2012)] The reviewer also seems to ignore completely the numerical validation of our model in our manuscript, which shows that f^4 is correct.
“Equation 6, there should be a function of the observer location r in the enhancement factor”	The field enhancement factor f does not depend on the observer position as can be seen from Eq. 12.69 in Novotny & Hecht. It only depends on the frequency ω. The observer location is built into the Green's function. $I(\mathbf{r}_\infty, \omega_R) = \frac{\omega_R^4}{\epsilon_0^4 c^4} \left [1 + f_2(\omega_R)] G_0(\mathbf{r}_\infty, \mathbf{r}_0) \alpha(\omega_R, \omega) [1 + f_1(\omega)] \right ^2 I_0(\mathbf{r}_0, \omega). \quad (12.69)$ [Novotny, L. & Hecht, B. Principles of nano-optics. (Cambridge University Press, 2012)]
“maybe only one factor of f contributes to the field enhancement, leading to f^2 as the maximal enhancement in absorption intensity, not f^4”	Our Eq. 6 and the f^4 scaling in intensity was proven both numerically and experimentally in several publications for Rayleigh (our case) & Raman scattering:  - Alonso-González, P. et al. Resolving the electromagnetic mechanism of surface-enhanced light scattering at single hot spots. Nat Commun 3, 684 (2012): $I_{AOA} \propto f ^4 \quad (5) \quad \text{-- shows } f^4 \text{ scaling}$ $E_{AOA} = G_A f \alpha_O f E_{inc} \quad (13) \quad \text{-- this is our Eq. 6}$  - Neuman, T. et al. Mapping the near fields of plasmonic nanoantennas by scattering-type scanning near-field optical microscopy: Mapping the near fields of plasmonic nanoantennas. Laser & Photonics Reviews 9, 637–649 (2015). - Le Ru, E. C. & Etchegoin, P. G. Rigorous justification of the $E ^4$ enhancement factor in Surface Enhanced Raman Spectroscopy. Chemical Physics Letters 423, 63–66 (2006).

	- Note that Reviewer #2 of this report also stresses the f^4 scaling “It turns out that the molecular scattering cross section appears as enhanced by the square (!) of the nearfield enhancement f in the scattering amplitude spectrum, which means that a measured intensity signal would be enhanced by the 4th power of f, like in SERS. FDTD simulation corroborate the results. This is an interesting result”
“The first enhancement factor is the local field enhancement factor, [...], the second enhancement factor is the enhanced molecular signal from the antenna to the detector [...] the two enhancement factors should not be the same.”	“Analysing the elastically scattered light ensures that the field enhancements f_1 and f_2 are exactly equal ($f_1 = f_2 = f$)”. [Alonso-González, P. et al. Resolving the electromagnetic mechanism of surface-enhanced light scattering at single hot spots. Nat Commun 3, 684 (2012).] “Using the reciprocity relation we arrive at $f'_{xx} = f_{xx}$ (29)“ where f_{xx} is the first enhancement factor and f'_{xx} the second enhancement factor. See chapter 2.2.1 in [Rezus, Y. L. A. & Selig, O. Impact of local-field effects on the plasmonic enhancement of vibrational signals by infrared nanoantennas. Opt. Express 24, 12202 (2016).]

Recommendation: This paper is not recommended because the physics is incorrect, and the research lacks a control experiment.

This is a very serious allegation, which the reviewer is unable to support in the provided comments. We find this statement unacceptable.

The experiment described is the detection of the absorption of the PDMS layer on an NSOM tip using a gold nanoparticle antenna to amplify the signal. Broad claims of a factor of 10^{13} enhancement are not supported in the manuscript. Control experiments are needed to verify several claims.

The reviewer is incorrect. The enhancement factors of 10^{13} are a result of our numerical calculations and are shown in Table 1 and Eqs. (17) and (18). Control experiments are generally not needed as near-field microscopy is able to provide background-free data. Nevertheless, in the revised version of our manuscript we now show control experiments that confirm our claims.

Experimental deficiency: Line 292 “Both near-field probe and antenna were illuminated from below in a transflection geometry with the weakly focused beam from a mid-infrared broadband laser (see Methods). Illumination from below ensured efficient excitation of the antenna while avoiding direct excitation of the near-field probe.” What is the control experiment to ensure that there is no measurement of the PDMA directly from the probe? How do we know that the detected light is only the scattered light from the nanoparticle antenna?

The reviewer is incorrect. The stated issue has been solved long ago in near-field microscopy, which is the basis for our experimental study. The answer is: we use demodulation of the detected light to suppress the direct scattering from the probe. In detail,

- It is well recognized that demodulation techniques are very effective at removing the direct tip scattering. See for example review papers Chen, X. et al. Modern Scattering-Type Scanning Near-Field Optical Microscopy for Advanced Material Research. Advanced Materials 31, 1804774 (2019).

- Our setup was described and validated in a prior publication (Virmani, D. et al. Amplitude- and Phase-Resolved Infrared Nanoimaging and Nanospectroscopy of Polaritons in a Liquid Environment. Nano Lett. 21, 1360–1367 (2021).
- Nevertheless, it is straightforward to show that direct tip-scattering is suppressed in our setup. To this end, we can compare the near-field signal on the rod antennas with the signal on the substrate (sufficiently away from the antenna to eliminate any coupling between antenna and tip).

In the revised manuscript, we now include a new Extended Data Fig. 6 and provide the following text in the Methods section.

In Extended Data Fig. 6(a) we show spectrally integrated near-field maps of the rod antennas. While the fundamental dipolar mode is observed on the rod antennas, as expected, importantly, the signal on the substrate is very small and below the noise floor. This was already observed for this setup in ref. 27. Line profiles taken across the long axis of the rod quantify this observation (Extended Data Fig. 6(b)). Performing near-field spectroscopy on the antenna and on the substrate further reveals that the near-field signal is near zero on the substrate (Extended Data Fig. 6(c)). Therefore, we can conclude that (i) demodulation suppresses any direct contribution of the PDMS-layer and (ii) the vibrational features observed in Fig. 3 solely stem from the molecules (on the tip) scattering via the antenna after being illuminated by the antenna, i.e. the AOA term.

In addition, the theory lacks rigor. The authors have failed to distinguish the enhancement factors that arose in two steps: the local field enhancement factor, which is the enhancement of the incident field broadcast by the antenna, and the object scattering enhancement factor, which is the enhancement of the infrared signal by the antenna. There are a few reasons to separate the two enhancement factors.

The reviewer is incorrect. The concept of field enhancement is well established in literature for the case where an object interacts with an antenna. For a general description, see refs.

- Alonso-González, P. et al. Resolving the electromagnetic mechanism of surface-enhanced light scattering at single hot spots. Nat Commun 3, 684 (2012):

- Neuman, T. et al. Mapping the near fields of plasmonic nanoantennas by scattering-type scanning near-field optical microscopy: Mapping the near fields of plasmonic nanoantennas. Laser & Photonics Reviews 9, 637–649 (2015).

1. Because the scattered field from the antenna is evanescent from the surface by localized surface plasmon, the enhancement factor f is a function of distance, which is not included in the theory of this paper. The first enhancement factor is the local field enhancement factor, and the local field is detected by the molecule, it is in near field; the second enhancement factor is the enhanced molecular signal from the antenna to the detector, which is in the far field. As a result, the two enhancement factors should not be the same. This also means that their maximum wavelength of enhancement is different significantly reducing any possible synergy.

The reviewer is incorrect. The field enhancement process is a near-field process. Further, first and second field enhancement factors are the same in SEIRA because of reciprocity. In other words, the illumination of the object by the antenna and the scattering of the object via the antenna are enhanced by the exact same factor. The concept of the field enhancement and its relevance for surface enhanced scattering is well established in literature:

- Rezus et al. provide a theoretical treatment of SEIRA. They clearly state that reciprocity applies and that the first and second field enhancement factors are equal in SEIRA. In more detail, they define the first enhancement factor as the field enhancement $f = E(\text{with antenna})/E(\text{no antenna})$ (Eq. 15 in Rezus). They define the second field enhancement factor as the dipole amplification factor f' (Eq. 17 in Rezus). To establish a relationship between f and f' , a test dipole is assumed (Fig. 3 in Rezus et al.). This procedure is shown to lead to a very clear result (chapter 2.2.1 in Rezus): both enhancement factors at the same owing the reciprocity, $f'_{xx} = f_{xx}$ (Eq. 29). This clearly contradicts the claims made by the reviewer that both should be different. [Rezus, Y. L. A. & Selig, O. Impact of local-field effects on the plasmonic enhancement of vibrational signals by infrared nanoantennas. Opt. Express 24, 12202 (2016).]
- We note that the enhancement factor f of course depends on the distance between the object and the antenna. This relation is taken into account in our paper. First, we evaluate f at the position of the object in the numerical calculation. Second, this dependence is the basis of our near-field experiment, where the tip is vibrated to vary f , thus allowing us to isolate the AOA term. The reviewer is incorrect in stating that this aspect is not considered.
- For clarity, here we derive an expression of the field enhancement, which is in agreement with Rezus et al. The AOA term in Eq. 6 of our paper can be written as a series of scattering events, treating the antenna and object as point objects.

$$\mathbf{E}_{\text{AOA}}(\mathbf{r}) = k^2 \mathbf{G}(\mathbf{r}, \mathbf{r}_A) \boldsymbol{\alpha}_A k^2 \mathbf{G}(\mathbf{r}_A, \mathbf{r}_O) \boldsymbol{\alpha}_O k^2 \mathbf{G}(\mathbf{r}_O, \mathbf{r}_A) \boldsymbol{\alpha}_A \mathbf{E}^{\text{in}}(\mathbf{r}_A)$$

We define the field enhancement tensor as follows (consistent with Rezus et al.)

$$\mathbf{f} = k^2 \mathbf{G}(\mathbf{r}_O, \mathbf{r}_A) \boldsymbol{\alpha}_A$$

From reciprocity it follows (see book “Dispersion Forces I by Yoshi Buhmann”, Appendix B, relation B.8) and assuming that the antenna polarizability is a diagonal matrix

$$\mathbf{f}^T = [k^2 \mathbf{G}(\mathbf{r}_O, \mathbf{r}_A) \boldsymbol{\alpha}_A]^T = k^2 \boldsymbol{\alpha}_A \mathbf{G}(\mathbf{r}_A, \mathbf{r}_O)$$

Where T is transpose. Inserting \mathbf{f} and \mathbf{f}^T in above Equation yields

$$\mathbf{E}_{\text{AOA}}(\mathbf{r}) = k^2 \mathbf{G}(\mathbf{r}, \mathbf{r}_A) \mathbf{f}^T \boldsymbol{\alpha}_O \mathbf{f} \mathbf{E}^{\text{in}}(\mathbf{r}_A)$$

By writing this Equation in scalar form with the approximation detailed in the Methods section, we obtain Eq. 6 of our paper. It follows that the electric field intensity, $|E_{\text{sca}}|^2$, scatters with f^4 . This result shows that the reviewer is incorrect.

- The reviewer may have confused inelastic light scattering (Raman) with elastic light scattering (IR). In Raman spectroscopy, the first and second field enhancement factors f_1 and f_2 are different because the Raman scattering occurs at a different frequency. See for example, Novotny, L. & Hecht, B. Principles of nano-optics. (Cambridge University Press, 2012). Here we point to the final result in the chapter “surface plasmons”, Eq. 12.70. It states that the total signal enhancement is given by $|[1 + f_2(\omega_R)] [1 + f_1(\omega)]|^2$ for a Raman scattering process. However, infrared spectroscopy (SEIRS) is an elastic light scattering process. In this case, scattering occurs at the same frequency as the illuminating field and hence, $f_1 = f_2$. This relation was demonstrated in numerical and near-field experiments. [Alonso-González, P. et al. Resolving the electromagnetic mechanism of surface-enhanced light scattering at single hot spots. Nat Commun 3, 684 (2012)]

To be clearer on the aspect of the field enhancement, we now add to page 5 of the main text f is the local field enhancement provided by the antenna **at the position of the object**

And provide the definition of f in the Methods section.

Note that the field enhancement tensor is defined as $\mathbf{f} = k^2 \mathbf{G}(\mathbf{r}_O, \mathbf{r}_A) \boldsymbol{\alpha}_A$.

2. The increase in local field can result in an increase in absorbance up to a saturation limit. Unlike the case of Raman scattering, where there can be many orders of magnitude of dynamic range in the enhancement, even absorption of 0.01 can only be increased by a circa two orders of magnitude before saturation occurs.

The principle assumption of SEIRA applications is that the object to be measured is extremely small, so much so, that it is virtually undetectable by far field methods alone. Examples are nanoparticles with molecular vibrations that have an extremely small absorption cross sections because of their small size. In these cases, enhancement factors of 5 to 6 orders of magnitude have been demonstrated in SEIRA. Our theory aims at describing this case.

3. Furthermore, the incident field is a plane wave, as defined in Equation 1. However, the scattered wave from the object is a spherical wave for an oscillating dipole, and it is from a much closer source for the antenna compared to the incident light. In this case, the geometric factor is different, the two enhancement factors may also be different in phase, which means, at one frequency, the first f reaches the maximum, the second one may be negligible. The molecular absorption frequency must have a relationship to the surface plasmon frequency in order for a resonance to be observed. This is not discussed.

The reviewer is incorrect.

- Both the antenna and the object (modelled as point polarizabilities) scatter light in form of a spherical wave. This is properly implemented by the Green's function, $\mathbf{G}(\mathbf{r}_O, \mathbf{r}_A)$, in the definition of the field enhancement, $\mathbf{f} = k^2 \mathbf{G}(\mathbf{r}_O, \mathbf{r}_A) \boldsymbol{\alpha}_A$. This description is consistent with Rezus et al. The reviewer is mistaken by writing that there is a fundamental difference between illuminating and scattering pathways. Both are the same, as demonstrated in Rezus et al and Neuman et al.

[Rezus, Y. L. A. & Selig, O. Impact of local-field effects on the plasmonic enhancement of vibrational signals by infrared nanoantennas. Opt. Express 24, 12202 (2016), Neuman, T. et al. Mapping the near fields of plasmonic nanoantennas by scattering-type scanning near-field optical microscopy: Mapping the near fields of plasmonic nanoantennas. Laser & Photonics Reviews 9, 637–649 (2015).]

- Aspects of the scattering of the antenna into the far field are included in the Green's function $\mathbf{G}(\mathbf{r}, \mathbf{r}_A)$:

$$\mathbf{E}_{AOA}(\mathbf{r}) = k^2 \mathbf{G}(\mathbf{r}, \mathbf{r}_A) \mathbf{f}^T \boldsymbol{\alpha}_O \mathbf{f} \mathbf{E}^{\text{in}}(\mathbf{r}_A)$$

Where r is the position of the detector and r_A the position of the antenna. We again stress that the position of the detector does not appear as an argument to the field enhancement f . Even when the position of the detector is changed ($\mathbf{G}(\mathbf{r}, \mathbf{r}_A)$ is changed), the scattered field intensity $|E_{AOA}|^2$ will still show a 4th-power dependence on f . This refutes the claim of the reviewer, that the geometric factor could have an influence on the f^4 scaling.

- The field enhancement is evaluated at the same frequency for both the illuminating light and scattered light. Further, from the definition of the field enhancement, $\mathbf{f} = k^2 \mathbf{G}(\mathbf{r}_O, \mathbf{r}_A) \boldsymbol{\alpha}_A$, and reciprocity, it follows that the phase of f is the same, contrary to what the reviewer writes.
- With above, it is now clear that the relationship between the molecular absorption frequency and the antenna resonance is fulfilled.

Therefore, in Equation 6, there should be a function of the observer location r in the enhancement factor, and maybe only one factor of f contributes to the field enhancement, leading to f^2 as the maximal enhancement in absorption intensity, not f^4 . Moreover, the phase of the antenna particle interaction may also be different further complicating the possible observations. There is no physical justification for the f^4 enhancement.

The reviewer is incorrect. The reviewer seems to be confused by a very important distinction. The scaling of the scattered field with the field enhancement is $|E_{sca}|^2 \sim |f|^4$. On the other hand, the molecular signature in the extinction cross section only scales with $\Delta\sigma^{ext} \sim |f|^2$. This is because of an interference effect. The discovery of this interference effect is one of the main results of our paper and discussed in detail. We speculate that the reviewer might have overlooked this important distinction. We further note:

- Our Equation 6 was proven both numerically and experimentally in Alonso-González, P. et al. Resolving the electromagnetic mechanism of surface-enhanced light scattering at single hot spots. Nat Commun 3, 684 (2012). See Eqs. (3) and (13) within that reference. It is thus wildly incorrect for the reviewer to state that there is no physical justification for the f^4 enhancement. Also, all our numerical simulations are confirming our Eq. 6 and our model in a general, which, however, the reviewer seems to completely ignore.
- Elastic light scattering of an object near an antenna is well established and is the foundation of our paper. For example, Alonso-Gonzales et al. describe it as a double scattering process and arrive at $|E_{sca}| \sim |f|^4$. Also Rezus et al. arrive at the same scaling. To see this, it is only necessary to expand p_{ant} in Eq. (6) in Rezus in the limit of small object polarizability, α_{vib} . The result is $p_{ant} \sim f^2$, and hence $|E_{sca}|^2 \sim |p_{ant}|^2 \sim |f|^4$. These and other literature, old and new, clearly support the physical consistency of the f^4 scaling:
 - Alonso-González, P. et al. Resolving the electromagnetic mechanism of surface-enhanced light scattering at single hot spots. Nat Commun 3, 684 (2012):
 - Neuman, T. et al. Mapping the near fields of plasmonic nanoantennas by scattering-type scanning near-field optical microscopy: Mapping the near fields of plasmonic nanoantennas. Laser & Photonics Reviews 9, 637–649 (2015).
 - Le Ru, E. C. & Etchegoin, P. G. Rigorous justification of the $|E|^4$ enhancement factor in Surface Enhanced Raman Spectroscopy. Chemical Physics Letters 423, 63–66 (2006).
 - Sun, J., Carney, P. S. & Schotland, J. C. Strong tip effects in near-field scanning optical tomography. Journal of Applied Physics 102, 103103 (2007).

Minor problems:

Line 68, please explain “the quantity” mentioned in “Current SEIRA models do not directly connect to the quantity that is measured in SEIRA”. The sentence is not clear.

We clarified this aspect and now write in the revised manuscript

Current SEIRA models do not directly connect to the quantity that is measured in SEIRA (i.e. whether it is absorption or scattering).

Line 131, the authors have mentioned the x direction. It is better to show the coordinate on Figure 1a to define the coordinate system.

Thank you for pointing this out. We have added the coordinate system.

Line 144, citation format for ref. 14 and 16 is different from others.

This find is well spotted, thank you. We corrected the format.

Line 245, in equations 17 and 18, is the background subtracted?

Yes, in all numerical calculations we subtract with a background simulation where the Lorentz-oscillator is switched off and only the background dielectric index of the object is considered, as detailed in the Methods. The reason is that we avoid the slight shifts in the antenna resonance induced by the background dielectric part of the object. This way, our resulting data correctly reproduces the vibrational line shapes and aligns more closely with the processing of experimental SEIRA where background in SEIRA spectra are fitted and subtracted.

Line 345, are there any experimental results for the control experiment?

Yes, we now provide results of control experiments, please see the answer to question 1.

Line 372, should be 1.8 ± 0.4 rather than 1.80 ± 0.40 , should use the error to round the sig figs.

Ok, we changed the format.

REVIEWER COMMENTS

Reviewer #1 (Remarks to the Author):

Actually, all of the comments and suggestions have been point-to-point responded by the authors. I suggest the manuscript be accepted now.

Reviewer #2 (Remarks to the Author):

In principle I am satisfied with the correction I recommended.

In order to satisfy referee 3, I recommend additional work on the explanations in the manuscript:

- Clear explanation, already in the title, that the paper studies the near-field nevertheless the detection the far-field.
- The quantity f is related to alpha-antenna. This relation should be explained shortly, at least for the antenna resonance condition. So, the text from line 203-217 could become better understandable and the strong enhancement in SEIRA scattering is better explained.
- It might also be useful to explain that for the near-field a spatial average is used (as far as I understand).
- Furthermore, I suggest to explain the Fano-line shapes in SEIRA, which are different for absorption and scattering.

Reviewer #3 (Remarks to the Author):

Please see attached document. Thanks.

[**Editorial note:** The document is at the end of these 4 pages.]

Reviewer #4 (Remarks to the Author):

The manuscript by Virmani et al. discusses the role of scattering in SEIRA by nanoantennas. The paper contains essentially two sections. The first one is the introduction of a model

based on interacting point like dipoles that enables to cast the extinction and the scattering cross sections as the sum of two terms, one due to the antenna and the other one accounting for the modification due to the presence of the vibrational modes.

The second section of the paper reports an experiment enabling to measure the scattered light with a clear signature of the vibrational modes. This experiment has already been published by the same group for a non-absorbing object. Here, the authors emphasize that the same technique can be used with an absorbing object.

In my view, the paper is technically correct. I find interesting the experimental section showing that the modulation of the distance separating the antenna from the object enables to obtain a good contrast. Thus, I recommend to publish the paper. However, the presentation must be changed. The authors must put forward the introduction of the new experimental modality based on the modulation of the object antenna distance but cannot claim any originality regarding the model and the discussion absorption versus scattering.

I do agree that many papers have used an oversimplified picture of absorption so that it is welcome to remind people that the signal is due to scattering. However, as they state themselves in the paper, optical theorem does not leave a choice: there is a scattered field that contains all the information. This simple remark could be sufficient to motivate the experiment and the modality suggested by the authors. The authors may keep this theoretical discussion for the sake of clarity but must remove all claims about originality as it closely follows Rezus and Selig work. Let me copy the conclusion of ref. 15 by Rezus and Selig which publish a very similar model of SEIRA in 2016:

"The model demonstrates that the main contribution to the cross-section change is due to a backaction process: the nanoantenna near field excites the molecular vibrations, which in turn radiate back onto the nanoantenna. This backaction process explains both the lineshape of the amplified vibrational signal and its amplitude. The lineshape is determined by the phase lag between the incident radiation and the reradiated field. Depending on the detuning between the nanoantenna and

the molecular vibrations, this phase lag can have any value between 0 and 2π , which explains the variety of observed lineshapes ranging from absorptive peaks and dips to fully dispersive shapes. The plasmonic enhancement of the vibrational signal is firstly determined by the square of the field enhancement of the nanoantenna."

In view of the conclusion of a paper published in 2016 (note that the word absorption is not written !!) , it seems clear that the following sentences of the abstract must be suppressed or strongly amended :

<-Here, we present an interpretation of SEIRA in form of a scattering process that identifies molecular scattering as the quantity that is measured in SEIRA.

-Molecular scattering – so far assumed to be negligible – is found to be enhanced...

It is shown that interferometric field-enhanced molecular scattering alone fully describes the magnitude and shape of vibrational lines in SEIRA extinction and scattering spectra.

>

The paper (and the abstract) should focus on the experimental part. The figure with the signal showing a 54% contrast should be in the main text.

Minor remarks :

1) "13 orders of magnitude" is self-explanatory and does not require the adjective "extraordinary".

2) The truncation in Eq. (2) may not be valid on resonance.

3) page 11. I do not understand the sentence:

Scanning the reference arm mirror (variation of E^{ref}) thus allowed us to suppress the interference term in the scattered field intensity, $2\text{Re}\{E_A^* \cdot E_{\text{AOA}}\}$ (Eq. (10)), and to measure directly E^{sca} in amplitude and phase. To further suppress the direct antenna scattering,

I expected to read $E_{\text{ref}} \cdot E_{\text{AOA}}$.

Scanning the mirror enables to perform the spectral analysis but cannot suppress the antenna contribution. Modulating the object-antenna distance enables to suppress the antenna direct contribution.

4) This observation provides clear evidence that the interaction between antenna and molecular vibration is a scattering process, and that field-enhanced molecular scattering can be measured and is a significant quantity.

Any far-field measurement is a measurement of a scattered field.

5) page 14:

for all three antennopas.

Editorial note: This is reviewer #3's document.

Recommendation: This paper does not meet the level of *Nature Communications* due to the manuscript's misinterpretation of concepts, lack of clarity, and compromises its accessibility to a diverse audience in *Nature Communications*.

Before I clarify my previous comments and their basis in the literature, it is worth pointing out fundamental contradictions in the manuscript that bears the title "Prediction and experimental verification of field-enhanced molecular scattering in SEIRA spectroscopy." SEIRA stands for *Surface Enhanced Infrared Absorption*. The technique being claimed in the manuscript involves scattering, which is distinct from absorption. For the definition and difference of scattering and absorption, please see Ref. [1] below. SEIRA is implemented most often as an evanescent wave spectroscopy like ATR-FTIR, but with the difference that the surface has a plasmon resonance that can be driven with IR electromagnetic radiation. The claim in this article that scattering by the "molecule" (a silica sphere) is as large as the absorption is based on a previous study Ref. [2].

The authors state on line 59 of the manuscript, "On the other hand, a recent publication showed that the spectral signature of molecular vibrations appears not only in the extinction cross section (which is typically measured in SEIRA experiments), but also in the scattering cross section of the antenna-molecule system" This statement is misrepresentation of what is observed. First, the previous paper was computational. Second, there were no computational controls. Third, what was demonstrated was linear combination of the absorption, extinction and scattering of a nanorod and the interfering absorbance by silica spheres. The only effect that could be considered due to plasmonic coupling was a larger absorbance of the silica along its transverse optical phonon band as the silica spheres are placed closer to the tips of the nanorod. This could be a SEIRA effect of perhaps one order of magnitude. There was no evidence of scattering from the silica.

There is no theoretical or experimental support for the state on line 65: "This description indicates that the spectral signature of molecular vibrations in the antenna scattering cross section could be explained by field-enhanced molecular scattering rather than field-enhanced molecular absorption." Since the scattering cannot be detected directly from the sample (a contaminant of PDMS in the case of this manuscript), the theory that supposedly explains the claim of a large scattering effect in SEIRA requires multiple scattering events between the molecule and nanorod. What is the evidence for such scattering events? The appearance of IR modes from the PDMS contaminant in Figure 3 are NSOM amplitude. But NSOM measures absorbance and reflectivity. It is entirely unclear how one would conclude that the signal of this unknown quantity of contaminant is a scattering signal that rivals the absorbance. Where is the control experiment for this point?

To say that "Current SEIRA models do not directly connect to the quantity that is measured in SEIRA (i.e. whether it is absorption or scattering)." is incorrect. There are direct measurements of infrared plasmons on IR plasmonic materials, such as doped metal oxides or the silver island films used by many groups showing the phase and intensity enhancement of the molecular absorption as it depends on the phase angle and frequency.

The issue of scattering also involves enhancement, which is also claimed in this manuscript. Enhancement has two stages, the first driven by a laser interacting with a nanoparticle, nanorod,

or nanostructure to drive a plasmon that amplifies the field experienced by the molecule. This enhancement depends on the square of the field and can affect absorption, fluorescence, and Raman. In Raman scattering the scattered photon has a different geometry and much less stored energy in the system since the exciting laser is at the Rayleigh frequency. The geometric effect is crucial [3,4], but is seldom mentioned. Even Rayleigh scattering must be correctly calculated based on the geometry of a scattering molecule near a nanoparticle or structure.

To say that the fourth power enhancement is settled science is to deny the active debate that exists. Etchegoin and Le Ru wrote a proof of E^4 [5] and yet they also wrote about the two stages of enhancement and pointed out that the truly large enhancements are due to Surface Enhanced Resonance Raman [6], which indicates the two stages of enhancement are different. These authors have not shown that there is a similar enhancement expected in both directions. They made a conjecture based on the reciprocity theorem, which is not well defined when resonance Raman enhancement is considered as a part of enhancement factor, and when nanoparticles are present along the path of light.

Please see the column on the right for clarification of our review comments and the references. This addition aims to highlight certain aspects that have been misinterpreted, and also to explain the rationale behind the points I've raised.

Reviewer #3 (author quoted)	Literature (author quoted)	Clarification
“There is no physical justification for the f^4 enhancement.”	Incredibly false statement and refuted by established literature including textbooks. “[...] we find that the Raman-scattered intensity scales linearly with the excitation intensity I_0 and that it depends on the factor” $\left [1 + f_2(\omega_R)][1 + f_1(\omega)] \right ^2.$ “Provided that $\omega_R \pm \omega$ is smaller than the spectral response of the metal nanostructure, the Raman-scattering enhancement scales roughly with the fourth power of the electric field enhancement.” “Notice that the theory outlined [above] is not specific to Raman scattering but applies also to any other linear interaction, such as Rayleigh scattering and fluorescence.” -- Rayleigh scattering is exactly the process that we study in our manuscript. [Novotny, L. & Hecht, B. Principles of nano-optics. (Cambridge University Press, 2012)] The reviewer also seems to ignore completely the numerical validation of our model in our manuscript, which shows that f^4 is correct.	The original comment is: “in Equation 6, there should be a function of the observer location r in the enhancement factor, and maybe only one factor of f contributes to the field enhancement, leading to f^2 as the maximal enhancement in absorption intensity, not f^4. Moreover, the phase of the antenna particle interaction may also be different further complicating the possible observations. There is no physical justification for the f^4 enhancement.” To the first quote, the Stokes shift is often broader than the spectral response of the nanosphere. With regard to the Rayleigh scattering comment, which Rayleigh scattering do the authors mean? The scattering is from the nanorod or structure not the PDMS contaminant. What is the mechanism whereby a scattered photon from a

		PDMS contaminant becomes 10⁶ photons are required by the statements made by the authors? The authors ignore the specific geometry of the molecule, which is not really a molecule, and the fact that the nature of molecule is not known since it has not been characterized. And, again there is no control experiment for reference.
“Equation 6, there should be a function of the observer location r in the enhancement factor”	The field enhancement factor f does not depend on the observer position as can be seen from Eq. 12.69 in Novotny & Hecht. It only depends on the frequency ω. The observer location is built into the Green’s function. $I(\mathbf{r}_\infty, \omega_R) = \frac{\omega_R^4}{\epsilon_0^2 c^4} \left [1 + f_2(\omega_R)] G_0(\mathbf{r}_\infty, \mathbf{r}_0) \alpha(\omega_R, \omega) [1 + f_1(\omega)] \right ^2 I_0(\mathbf{r}_0, \omega).$ [Novotny, L. & Hecht, B. Principles of nano-optics. (Cambridge University Press, 2012)]	In the referenced work [7], there was no specific mention of whether the factor f depends on the observer’s position or the position of the volume of interest. The level of theory is quite general, but surely the authors are aware that a Green’s function depends on geometry and scattering intensity depends on the orientation of the molecule relative to the nanostructure. Mie, Raman and Raleigh scattering depend on the position of the detector. Why do these authors believe that they have a way to defy physical principles?
“maybe only one factor of f contributes to the field enhancement, leading to f^2 as the maximal enhancement in absorption intensity, not f^4”	Our Eq. 6 and the f^4 scaling in intensity was proven both numerically and experimentally in several publications for Rayleigh (our case) & Raman scattering: - Alonso-González, P. et al. Resolving the electromagnetic mechanism of surface-enhanced light scattering at single hot spots. Nat Commun 3, 684 (2012): $I_{AOA} \propto f ^4 \quad (5) \quad \text{-- shows } f^4 \text{ scaling}$ $E_{AOA} = G_A f \alpha_O f E_{inc} \quad (13) \quad \text{-- this is our Eq. 6}$ - Neuman, T. et al. Mapping the near fields of plasmonic nanoantennas by scattering-type scanning near-field optical microscopy: Mapping the near fields of plasmonic nanoantennas. Laser & Photonics Reviews 9, 637–649 (2015). - Le Ru, E. C. & Etchegoin, P. G. Rigorous justification of the E ⁴ enhancement factor in Surface Enhanced	It’s true that some literature does indicate an f^4 enhancement, but it is important to note that these findings are often based on specific assumptions. In Equation 6 in this manuscript, my discussion is focused on the scenario where the two enhancements differ.

	Raman Spectroscopy. Chemical Physics Letters 423, 63–66 (2006). - Note that Reviewer #2 of this report also stresses the f^4 scaling “It turns out that the molecular scattering cross section appears as enhanced by the square (!) of the nearfield enhancement f in the scattering amplitude spectrum, which means that a measured intensity signal would be enhanced by the 4th power of f, like in SERS. FDTD simulation corroborate the results. This is an interesting result”	
“The first enhancement factor is the local field enhancement factor, [...], the second enhancement factor is the enhanced molecular signal from the antenna to the detector [...] the two enhancement factors should not be the same.”	“Analysing the elastically scattered light ensures that the field enhancements f_1 and f_2 are exactly equal ($f_1 = f_2 = f$)”. [Alonso-González, P. et al. Resolving the electromagnetic mechanism of surface-enhanced light scattering at single hot spots. Nat Commun 3, 684 (2012).] “Using the reciprocity relation we arrive at $f'_{xx} = f_{xx}$ (29)“ where f_{xx} is the first enhancement factor and f'_{xx} the second enhancement factor. See chapter 2.2.1 in [Rezus, Y. L. A. & Selig, O. Impact of local-field effects on the plasmonic enhancement of vibrational signals by infrared nanoantennas. Opt. Express 24, 12202 (2016).]	The author’s previous paper [8] stated that $f_1=f_2=f$, however, it’s important to recognize that this statement is based on an assumption detailed in the original paper (on page 2), specifically that the illumination and detection directions are the same, a consequence of the reciprocity theorem. For a comprehensive understanding, it would be beneficial if this assumption were clearly outlined in this manuscript. Furthermore, this assumption underscores that the enhancement factor is influenced by the position, leading to the possibility that the two enhancement factors might not be equal in most scenarios. Regarding the application of the reciprocity theorem, as originally discussed in the work [5], it’s a pivotal concept that the enhancement factor is a function of position. However, it has less emphasis the nanoparticle. This may suggest a need for a more nuanced application of this application of the reciprocity theorem.

The above references provide evidence that the enhancement factor is influenced by position, suggesting that the two enhancement factors are not equal in most cases. These aspects would

require further explanation and discussion in the manuscript to provide a clearer understanding of the underlying principles and their application in this context.

Labeling differing scientific views as “unacceptable” or “incredibly false” is not a professional approach. The literature in this field is full of contradictory viewpoints and this is not settled science. The annals of scientific history are replete with such instances where debate is needed for progress. Through open discussion, debate, and effective communication, we refine our understanding. By designing well-structured experiments and theories, we create conditions that are clearly defined and conducive to objective analysis. This methodical approach fosters a more inclusive and productive scientific discourse.

In summary, the rationale behind my recommendation not to publish in *Nature Communications* lies in the manuscript has failed on the two points:

1. Given the diverse readership of *Nature Communications*, clarity in presenting theories and basic assumptions is critical. If these elements are unclear or absent, the results could be misconstrued. Authors should write with the understanding that their audience may not specialize in their field and should value diverse feedback. Improving the manuscript’s accessibility should be a priority over voicing grievances about audience feedback being “incredibly false”. It’s important to clearly outline assumptions for all readers, even for those who are familiar with the research.
2. From the perspective of a chemist and physicist, adhering to fundamental principles of scholarly work, such as accurate interpretation and citation of literature, is essential. Misquoting or misrepresenting sources not only diminishes the author’s argument but also jeopardizes the scientific discourse’s integrity. A thorough review of the citations is required to ensure their correct and appropriate usage.

References:

- [1] Jackson, J. D. *Classical electrodynamics* (John Wiley & Sons, 1962).
- [2] Neuman, T. et al. Importance of Plasmonic Scattering for an Optimal Enhancement of Vibrational Absorption in SEIRA with Linear Metallic Antennas. *J. Phys. Chem. C* **119**, 26652–26662 (2015).
- [3] Moskovits, M. Surface-enhanced spectroscopy. *Rev. Mod. Phys.* **57**, 783 (1985).
- [4] Etchegoin, P. G. et al. Polarization-dependent effects in surface-enhanced Raman scattering (SERS) *Phys. Chem. Chem. Phys.* **8**, 2624 (2006).
- [5] Le Ru, E. C. & Etchegoin, P. G. Rigorous justification of the $|E|^4$ enhancement factor in Surface Enhanced Raman Spectroscopy. *Chem. Phys. Lett.* **423**, 63–66 (2006)
- [6] Le Ru, E. C. et al. Surface Enhanced Raman Scattering Enhancement Factors: A Comprehensive Study. *Phys. Chem. C* **111**, 13794 (2007).
- [7] Novotny, L. & Hecht, B. *Principles of nano-optics* (Cambridge University Press, 2012)
- [8] Alonso-González, P. et al. *Nat. Commun.* **3**, 684 (2012)

Response to Reviewer 1:

Actually, all of the comments and suggestions have been point-to-point responded by the authors. I suggest the manuscript be accepted now.

We thank the reviewer for the positive evaluation and the recommendation to accept our manuscript.

Response to Reviewer 2:

In principle I am satisfied with the correction I recommended..

We thank the reviewer for the positive evaluation and the recommendation to accept our manuscript.

In order to satisfy referee 3, I recommend additional work on the explanations in the manuscript:

- Clear explanation, already in the title, that the paper studies the near-field nevertheless the detection the far-field.

This might not have been the problematic point because s-SNOM is well known as a near-field experiment with detection in the far field, and this was not challenged by the reviewer #3. We do now clarify one important detail concerning the far-field detection, that is, that the illumination and scattering directions are the same, thus ensuring equal field enhancement and a f^4 scaling law.

Page 9

As in ref.¹⁷, illumination and detection directions are the same, and as a consequence of reciprocity, it is thus ensured that the field enhancement of the illumination and scattering pathways are equal.

-The quantity f is related to alpha-antenna. This relation should be explained shortly, at least for the antenna resonance condition. So, the text from line 203-217 could become better understandable and the strong enhancement in SEIRA scattering is better explained.

We state the relation between f and α_{antenna} for the resonance case on page 8 to show more clearly how Eq. (14) was derived.

Page 8

In the case where the antenna resonance is tuned to the molecular vibration (that is, $\text{Arg}(f) = \pi/2$ and also $\text{Arg}(\alpha_A) = \pi/2$ because of $f = k^2 G_{xx}(\mathbf{r}_O, \mathbf{r}_A) \alpha_A$), Eq. (13) becomes [...].

-It might also be useful to explain that for the near-field a spatial average is used (as far as I understand).

The field enhancement is evaluated at the center of the small particle (O). We only need to compute the spatial average for large particles (30 nm radius). We now write this more clearly in the Methods section on page 21.

Note that for NP radius of 10 nm or lower, it is sufficient to evaluate the field enhancement at the center of the NP. Only in case of NP radius of 30 nm, where the inhomogeneity of the antenna near fields become significant, the average field enhancement factor $f_{\text{avg}}^2 = \frac{1}{V_{\text{np}}} \int_{V_{\text{np}}} f^2 dV$ was calculated.¹⁵

- Furthermore, I suggest to explain the Fano-line shapes in SEIRA, which are different for absorption and scattering.

This is indeed a great idea. We briefly discuss the different line shapes in extinction and scattering cross section in the theory part and in the numerical study.

Page 7

The spectral signature of the molecular vibration in extinction cross section (Fig. 1(c)) is determined by the interference of the incident field \mathbf{E}^{in} with the field-enhanced molecular scattering \mathbf{E}_{AOA} (the 2nd term in Eq. (9)). In case of the scattering cross section (Eq.(10)), the spectral signature of the molecular vibration is caused by the interference of field-enhanced molecular scattering \mathbf{E}_{AOA} with the direct antenna scattering \mathbf{E}_{A} rather than the incident field (the 2nd term in Eq. (10)), explaining the differences in the magnitude and shape of the vibrational line shapes (Fig. 1(c), Extended Data Fig. 2).

Page 9

Briefly, antennas driven far below resonance (short antennas, $\text{Arg}(f) \rightarrow 0$) yield a spectral signature of the molecular vibration in form of a peak in the extinction cross section ($\text{Im}\{\alpha_0\}$). Antennas tuned to the molecular resonance ($\text{Arg}(f) = \pi/2$) yield a dip ($-\text{Im}\{\alpha_0\}$) and antennas driven far above resonance (long antennas, $\text{Arg}(f) \rightarrow \pi$) yield again a peak ($\text{Im}\{\alpha_0\}$). Because of the different interference process (cf. Eqs. (9), (10)), the vibrational line shape in the scattering cross section follows the sequence from normal dispersive ($\text{Re}\{\alpha_0\}$) to a dip ($-\text{Im}\{\alpha_0\}$) to anomalous dispersive ($-\text{Re}\{\alpha_0\}$) as the antenna length is varied from short to tuned to long.

Response to Reviewer 3:

We thank the reviewer for the time to evaluate our manuscript and input to improve our manuscript.

1) Recommendation: This paper does not meet the level of Nature Communications due to the manuscript's misinterpretation of concepts, lack of clarity, and compromises its accessibility to a diverse audience in Nature Communications.

Before I clarify my previous comments and their basis in the literature, it is worth pointing out fundamental contradictions in the manuscript that bears the title "Prediction and experimental verification of field-enhanced molecular scattering in SEIRA spectroscopy." SEIRA stands for Surface Enhanced Infrared Absorption. The technique being claimed in the manuscript involves scattering, which is distinct from absorption. For the definition and difference of scattering and absorption, please see Ref. [1] below. SEIRA is implemented most often as an evanescent wave spectroscopy like ATR-FTIR, but with the difference that the surface has a plasmon resonance that can be driven with IR electromagnetic radiation. The claim in this article that scattering by the "molecule" (a silica sphere) is as large as the absorption is based on a previous study Ref. [2].

What we meant by SEIRA was the technique, not the process. We agree that our original wording was misleading and we now connect our work differently to SEIRA. In this regard, the following changes were made to the manuscript:

- (i) We clarify throughout our manuscript that we consider a very specific SEIRA system: molecular vibrations couple to a single IR-resonant antenna. This system was considered in the first experimental demonstration of resonant SEIRA and is often studied because of its simplicity and generality.

Page 4:

Here we explore the role of molecular scattering when mid-infrared molecular vibrations are located in the vicinity of a single antenna – a system that is usually considered in SEIRA experiments. We developed and numerically verified an analytical model that describes the vibrational line shapes in the extinction cross section as the interference between field-enhanced molecular scattering and the incident field.

- (ii) We have also reworded the paragraph "So far, it is widely accepted that..." in the introduction where we have removed the scattering vs absorption argumentation, and instead motivate our study by citing literature that indicates the relevance of molecular scattering when molecular vibrations couple to a single antenna.

Page 3:

On the other hand, Alonso-Gonzalez et al. verified experimentally that the infrared elastic light scattering from a small non-absorbing particle in the vicinity of a single antenna is analogous to the mechanism of surface-enhanced Raman spectroscopy (SERS), that is, the intensity of the infrared elastic light scattering from the particle is enhanced by the fourth power of the local field enhancement, $|f|^4$.¹⁷ The underlying process was described as scattering from the particle via the antenna after being illuminated by the antenna.^{18,19} Rezus and Selig then assumed a small particle exhibiting molecular vibrations in the vicinity of the antenna and showed in a numerical study that field-enhanced molecular scattering fully describes the line shapes in extinction spectra of the antenna-particle system.¹⁵ Experimental verification of field-enhanced molecular scattering as well as its connection with the vibrational line shapes is, however, challenging because of the difficulty to measure the scattered light from the antenna free of any interference with the incident field.

- (iii) We clarified in the theory section that we develop the expressions for the molecular line shapes for the molecular-single antenna system, rather than for SEIRA in general. We

connect with SEIRA by showing that our scattering model yields the expression for field-enhanced molecular absorption, in consistency with Rezus and Selig.

Page 8:

At antenna resonance, the spectral signature of the molecular vibration in the extinction cross section, $\sigma_{\text{vib}}^{\text{ext}}$, based on an exclusively scattering model is formally identical with field-enhanced molecular absorption, $k|f|^2\text{Im}\{\alpha_0\}$, which is the typical accepted explanation of SEIRA. Thus, the vibrational signal in typical SEIRA spectra of single resonant antennas can be equivalently well understood as field-enhanced molecular absorption or interferometrically- and field-enhanced molecular scattering.

2) The authors state on line 59 of the manuscript, “On the other hand, a recent publication showed that the spectral signature of molecular vibrations appears not only in the extinction cross section (which is typically measured in SEIRA experiments), but also in the scattering cross section of the antenna-molecule system” This statement is misrepresentation of what is observed. First, the previous paper was computational. Second, there were no computational controls. Third, what was demonstrated was linear combination of the absorption, extinction and scattering of a nanorod and the interfering absorbance by silica spheres. The only effect that could be considered due to plasmonic coupling was a larger absorbance of the silica along its transverse optical phonon band as the silica spheres are placed closer to the tips of the nanorod. This could be a SEIRA effect of perhaps one order of magnitude. There was no evidence of scattering from the silica.

This paper (Neuman et al.) demonstrated plasmonic scattering in SEIRA, and thus we cite it. Following the comment from the reviewer, we do not bring it in connection with molecular scattering anymore. We note, however, that the paper of Rezus and Selig *Opt. Express* **24**, 12202 (2016) showed that molecular scattering can predict the molecular line shapes in the extinction cross section when considering a single antenna, which we now corroborate in our experiment.

3) There is no theoretical or experimental support for the state on line 65: “This description indicates that the spectral signature of molecular vibrations in the antenna scattering cross section could be explained by field-enhanced molecular scattering rather than field-enhanced molecular absorption.”

We removed this statement.

4) Since the scattering cannot be detected directly from the sample (a contaminant of PDMS in the case of this manuscript), the theory that supposedly explains the claim of a large scattering effect in SEIRA requires multiple scattering events between the molecule and nanorod. What is the evidence for such scattering events? The appearance of IR modes from the PDMS contaminant in Figure 3 are NSOM amplitude. But NSOM measures absorbance and reflectivity. It is entirely unclear how one would conclude that the signal of this unknown quantity of contaminant is a scattering signal that rivals the absorbance. Where is the control experiment for this point?

s-SNOM is a pure scattering experiment, where the scattered light from the near-field probe is detected. Typically, the amplitude of the scattered light can be related to the sample reflectivity and the phase to sample absorption, but only when a metal tip is used and planar samples are assumed. This is, however, not what we do in our manuscript, so we cannot make this comparison. Instead, we have implemented a specific s-SNOM experiment where we detect the scattered light from the near-field probe via a single antenna, after being illuminated by the antenna. This process was shown by Alonso-Gonzalez et al *Nat. Commun.* **3**, 684 (2012) to be a pure scattering experiment, which was further proven by us by finding the f^4 power law in our data in Fig. 3.

In the revised manuscript, we now use our experimental s-SNOM data, which yields a signal proportional to field-enhanced molecular scattering, and show that we can mimic the spectral line shapes

in the extinction spectra by a proper projection of the s-SNOM data. We thus confirm the prediction by Rezus and Selig *Opt. Express* **24**, 12202 (2016). To this end, we introduced a new Fig 4 and added the corresponding text on page 16:

In Fig. 4 we establish a connection between the near-field experiment shown in Fig. 2 with the vibrational line shapes observed in the extinction cross section. To this end, we plot the baseline-corrected extinction cross section, $\Delta\sigma^{\text{ext}} = \sigma^{\text{ext}} - \sigma_{\text{bkg}}^{\text{ext}}$, in Fig. 4(a), where σ^{ext} and $\sigma_{\text{bkg}}^{\text{ext}}$ is the extinction cross section for an absorbing and a non-absorbing object, respectively. This baseline correction effectively removes the direct antenna scattering from σ^{ext} (\mathbf{E}_A in Eq. (9)) and thus isolates the spectral signature of the molecular vibration. For comparison, we plot the imaginary part of the baseline-corrected near-field spectra, $\Delta\tilde{\epsilon}_3 = \tilde{\epsilon}_3 - \tilde{\epsilon}_3^{\text{bkg}}$, in Fig. 4(b), where $\tilde{\epsilon}_3$ is the measured near-field spectrum (solid lines in Fig. 3(a,d)) and $\tilde{\epsilon}_3^{\text{bkg}}$ is the calculated near-field spectrum assuming a near-field probe that is covered by non-absorbing dielectric thin film (shaded area in Fig. 3(a)). This baseline correction is motivated as follows: While the direct antenna scattering \mathbf{E}_A is the reason for the baseline effect in the extinction cross section, \mathbf{E}_A is suppressed in the near-field experiment by the demodulation techniques. Instead, it is the field-enhanced scattering from the metal core of the near-field probe that produces a strong baseline effect in the s-SNOM experiment, yielding the antenna resonance curve (the shaded area in Fig 3(a)). We note that the polarizability of the PDMS-covered near-field probe may be separated into the sum of background (describing essentially the metal core) and vibrational components (describing the molecular vibration), which can be shown by developing Eq. (36) in the limit of a weak molecular oscillator. Further noting that $\tilde{\epsilon}_3 \propto \mathbf{E}_{\text{AOA}} \propto \alpha_O$ (Eqs. (6), (17)), the field-enhanced molecular scattering can be isolated by taking the complex-valued difference, $\Delta\tilde{\epsilon}_3 = \tilde{\epsilon}_3 - \tilde{\epsilon}_3^{\text{bkg}}$. The imaginary part of $\Delta\tilde{\epsilon}_3$ can then be calculated straightforwardly, which is motivated by Eq. (9), and possible because the antenna-scattered light is resolved in amplitude and phase and normalized to the incident field \mathbf{E}_{in} (Methods). The result is a good agreement of the s-SNOM data with the vibrational line shape in the extinction cross section $\Delta\sigma^{\text{ext}}$ (cf. Fig. 4(a,b)), thus providing experimental evidence that the interference between field-enhanced molecular scattering \mathbf{E}_{AOA} and the incident field can fully explain the vibrational signature in SEIRA spectra.

5) To say that “Current SEIRA models do not directly connect to the quantity that is measured in SEIRA (i.e. whether it is absorption or scattering).” is incorrect. There are direct measurements of infrared plasmons on IR plasmonic materials, such as doped metal oxides or the silver island films used by many groups showing the phase and intensity enhancement of the molecular absorption as it depends on the phase angle and frequency.

Although our statement is true, we remove it as it is not important.

6) The issue of scattering also involves enhancement, which is also claimed in this manuscript. Enhancement has two stages, the first driven by a laser interacting with a nanoparticle, nanorod, or nanostructure to drive a plasmon that amplifies the field experienced by the molecule. This enhancement depends on the square of the field and can affect absorption, fluorescence, and Raman. In Raman scattering the scattered photon has a different geometry and much less stored energy in the system since the exciting laser is at the Rayleigh frequency. The geometric effect is crucial [3,4], but is seldom mentioned. Even Rayleigh scattering must be correctly calculated based on the geometry of a scattering molecule near a nanoparticle or structure.

We note that we do not consider Raman spectroscopy and we stress that we exclusively consider Rayleigh scattering. In the revised manuscript, we point out that the illumination and detection directions are the same, and thus the orientation of the molecule does not matter because of reciprocity.

As a side note, we do not consider a single molecule (which would have a specific orientation), but a nanoparticle of randomly oriented molecules (thus justifying the assumption of an isotropic polarizability of the object). We clarify this aspect in the text on Page 20:

Third, the polarizability of the spherical object is assumed to be isotropic ($\alpha_{0,xx} = \alpha_{0,yy} = \alpha_{0,zz}$), that is, we consider that the object consists of a large number of randomly oriented molecules.

To say that the fourth power enhancement is settled in science is to deny the active debate that exists. Etchegoin and Le Ru wrote a proof of EF^4 [5] and yet they also wrote about the two stages of enhancement and pointed out that the truly large enhancements are due to Surface Enhanced Resonance Raman [6], which indicates the two stages of enhancement are different. These authors have not shown that there is a similar enhancement expected in both directions. They made a conjecture based on the reciprocity theorem, which is not well defined when resonance Raman enhancement is considered as a part of enhancement factor, and when nanoparticles are present along the path of light.

We agree that in Raman additional effects are relevant, but we are not aware that they are relevant in SEIRA because resonance shifts do not play a role for example. Indeed, Rezus and Selig *Opt. Express* **24**, 12202 (2016) showed that the reciprocity theorem applies when considering same illumination and detection direction, which is our case.

7) To the first quote, the Stokes shift is often broader than the spectral response of the nanosphere. With regard to the Rayleigh scattering comment, which Rayleigh scattering do the authors mean? The scattering is from the nanorod or structure not the PDMS contaminant. What is the mechanism whereby a scattered photon from a PDMS contaminant becomes 10^6 photons are required by the statements made by the authors? The authors ignore the specific geometry of the molecule, which is not really a molecule, and the fact that the nature of molecule is not known since it has not been characterized. And, again there is no control experiment for reference.

The mechanism is the superposition of the scattered light from the PDMS-coated near-field probe with the reference beam in s-SNOM, which provides an interferometric gain. For reference, this effect was described for a standard s-SNOM experiment in F. Huth et al. *Nature Mater* **10**, 352–356 (2011). When considering the coupling of molecular vibrations to a single antenna, we make the same assumptions as in Rezus and Selig *Opt. Express* **24**, 12202 (2016), where the specific geometry of the molecule is not relevant.

8) In the referenced work [7], there was no specific mention of whether the factor f depends on the observer's position or the position of the volume of interest. The level of theory is quite general, but surely the authors are aware that a Green's function depends on geometry and scattering intensity depends on the orientation of the molecule relative to the nanostructure. Mie, Raman and Raleigh scattering depend on the position of the detector. Why do these authors believe that they have a way to defy physical principles?

The field enhancement may change with the observer's position but not the scaling law when the antenna is approximated as a point dipole. The scaling law is the relevant quantity in our study, not the absolute value of the field enhancement. Further, the illumination and detection direction are the same in our experiment, so the reviewer's consideration does not apply to our case.

9) It's true that some literature does indicate an f^4 enhancement, but it is important to note that these findings are often based on specific assumptions. In Equation 6 in this manuscript, my discussion is focused on the scenario where the two enhancements differ.

The author's previous paper [8] stated that $f_1=f_2=f$, however, it's important to recognize that this statement is based on an assumption detailed in the original paper (on page 2), specifically that the illumination and detection directions are the same, a consequence of the reciprocity theorem. For a comprehensive understanding, it would be beneficial if this assumption were clearly outlined in this manuscript.

Furthermore, this assumption underscores that the enhancement factor is influenced by the position, leading to the possibility that the two enhancement factors might not be equal in most scenarios.

Regarding the application of the reciprocity theorem, as originally discussed in the work [5], it's a pivotal concept that the enhancement factor is a function of position. However, it has less emphasis on the nanoparticle. This may suggest a need for a more nuanced application of this application of the reciprocity theorem.

The reviewer now acknowledges that the f^4 scaling law is possible under certain assumptions. Our experiment assumes same illumination and detection directions, thus corresponding field enhancements are equal, $f_1=f_2$, and reciprocity holds. This was stated in the original manuscript, but we now write this even clearer (see answer to point 6).

Response to Reviewer 4:

We thank the reviewer for the positive evaluation and comments to improve our manuscript.

The manuscript by Virmani et al. discusses the role of scattering in SEIRA by nanoantennas. The paper contains essentially two sections. The first one is the introduction of a model based on interacting point like dipoles that enables to cast the extinction and the scattering cross sections as the sum of two terms, one due to the antenna and the other one accounting for the modification due to the presence of the vibrational modes.

The second section of the paper reports an experiment enabling to measure the scattered light with a clear signature of the vibrational modes. This experiment has already been published by the same group for a non-absorbing object. Here, the authors emphasize that the same technique can be used with an absorbing object.

In my view, the paper is technically correct. I find interesting the experimental section showing that the modulation of the distance separating the antenna from the object enables to obtain a good contrast. Thus, I recommend to publish the paper. However, the presentation must be changed. The authors must put forward the introduction of the new experimental modality based on the modulation of the object antenna distance but cannot claim any originality regarding the model and the discussion absorption versus scattering.

I do agree that many papers have used an oversimplified picture of absorption so that it is welcome to remind people that the signal is due to scattering. However, as they state themselves in the paper, optical theorem does not leave a choice: there is a scattered field that contains all the information. This simple remark could be sufficient to motivate the experiment and the modality suggested by the authors. The authors may keep this theoretical discussion for the sake of clarity but must remove all claims about originality as it closely follows Rezus and Selig work. Let me copy the conclusion of ref. 15 by Rezus and Selig which publish a very similar model of SEIRA in 2016:

"The model demonstrates that the main contribution to the cross-section change is due to a backaction process: the nanoantenna near field excites the molecular vibrations, which in turn radiate back onto the nanoantenna. This backaction process explains both the lineshape of the amplified vibrational signal and its amplitude. The lineshape is determined by the phase lag between the incident radiation and the reradiated field. Depending on the detuning between the nanoantenna and the molecular vibrations, this phase lag can have any value between 0 and 2π , which explains the variety of observed lineshapes ranging from absorptive peaks and dips to fully dispersive shapes. The plasmonic enhancement of the vibrational signal is firstly determined by the square of the field enhancement of the nanoantenna."

In view of the conclusion of a paper published in 2016 (note that the word absorption is not written !!) , it seems clear that the following sentences of the abstract must be suppressed or strongly amended :

<-Here, we present an interpretation of SEIRA in form of a scattering process that identifies molecular scattering as the quantity that is measured in SEIRA.

-Molecular scattering – so far assumed to be negligible – is found to be enhanced...

It is shown that interferometric field-enhanced molecular scattering alone fully describes the magnitude and shape of vibrational lines in SEIRA extinction and scattering spectra.>

The paper (and the abstract) should focus on the experimental part. The figure with the signal showing a 54% contrast should be in the main text.

We thank the reviewer for the recommendation to publish our manuscript and for recognizing the importance of scattering in the interpretation of SEIRA. We follow the suggestions of the reviewer and implemented the following changes:

- (i) We now highlight more the experimental section. To this end, we have introduced a new Fig 4 where we show the connection of the s-SNOM experiment with SEIRA spectroscopy. We have further moved the figure on the gap antenna showing the 54% molecular contrast to the main text as a new Fig 6, as requested.

Page 15

In Fig. 4 we establish a connection between the near-field experiment shown in Fig. 2 with the vibrational line shapes observed in the extinction cross section. To this end, we plot the baseline-corrected extinction cross section, $\Delta\sigma^{\text{ext}} = \sigma^{\text{ext}} - \sigma_{\text{bkg}}^{\text{ext}}$, in Fig. 4(a), where σ^{ext} and $\sigma_{\text{bkg}}^{\text{ext}}$ is the extinction cross section for an absorbing and a non-absorbing object, respectively. This baseline correction effectively removes the direct antenna scattering from σ^{ext} (\mathbf{E}_A in Eq. (9)) and thus isolates the spectral signature of the molecular vibration. For comparison, we plot the imaginary part of the baseline-corrected near-field spectra, $\Delta\tilde{\epsilon}_3 = \tilde{\epsilon}_3 - \tilde{\epsilon}_3^{\text{bkg}}$, in Fig. 4(b), where $\tilde{\epsilon}_3$ is the measured near-field spectrum (solid lines in Fig. 3(a,d)) and $\tilde{\epsilon}_3^{\text{bkg}}$ is the calculated near-field spectrum assuming a near-field probe that is covered by non-absorbing dielectric thin film (shaded area in Fig. 3(a)). This baseline correction is motivated as follows: While the direct antenna scattering \mathbf{E}_A is the reason for the baseline effect in the extinction cross section, \mathbf{E}_A is suppressed in the near-field experiment by the demodulation techniques. Instead, it is the field-enhanced scattering from the metal core of the near-field probe that produces a strong baseline effect in the s-SNOM experiment, yielding the antenna resonance curve (the shaded area in Fig 3(a)). We note that the polarizability of the PDMS-covered near-field probe may be separated into the sum of background (describing essentially the metal core) and vibrational components (describing the molecular vibration), which can be shown by developing Eq. (36) in the limit of a weak molecular oscillator. Further noting that $\tilde{\epsilon}_3 \propto \mathbf{E}_{\text{AOA}} \propto \alpha_0$ (Eqs. (6), (17)), the field-enhanced molecular scattering can be isolated by taking the complex-valued difference, $\Delta\tilde{\epsilon}_3 = \tilde{\epsilon}_3 - \tilde{\epsilon}_3^{\text{bkg}}$. The imaginary part of $\Delta\tilde{\epsilon}_3$ can then be calculated straightforwardly, which is motivated by Eq. (9), and possible because the antenna-scattered light is resolved in amplitude and phase and normalized to the incident field \mathbf{E}_{in} (Methods). The result is a good agreement of the s-SNOM data with the vibrational line shape in the extinction cross section $\Delta\sigma^{\text{ext}}$ (cf. Fig. 4(a,b)), thus providing experimental evidence that the interference between field-enhanced molecular scattering \mathbf{E}_{AOA} and the incident field can fully explain the vibrational signature in SEIRA spectra.

- (ii) We have reworked the theory section of this manuscript by writing it more as a reminder as a discovery. To this end, discussions were shortened and Table 1 was moved to the Extended Data, reducing this section from 9 to 5 pages. However, we note that Rezus and Selig only discuss the SEIRA line shapes as they appear in the extinction cross section. Thus, the following two aspects were not previously discussed and were kept in our theory section: (a) the interference process between the field-enhanced molecular scattering and the incident field and the importance of the resulting interferometric gain and (b) the expressions for the molecular line shapes in the scattering and absorption cross section, thus providing this complete description. We end the theory section by connecting the presented scattering model with SEIRA and by showing that our scattering model yields the expression for field-enhanced molecular scattering, in consistency with Rezus and Selig.

Page 8:

At antenna resonance, the spectral signature of the molecular vibration in the extinction cross section, $\sigma_{\text{vib}}^{\text{ext}}$, based on an exclusively scattering model is formally identical with field-enhanced molecular absorption, $k|f|^2\text{Im}\{\alpha_0\}$, which is the typical accepted explanation of SEIRA. Thus, the vibrational signal in typical SEIRA spectra of single resonant antennas can be equivalently well understood as field-enhanced molecular absorption or interferometrically- and field-enhanced molecular scattering.

- (iii) As a result of above changes, we have removed the word “prediction” from our title and have rewritten the abstract.

Minor remarks :

1) "13 orders of magnitude" is self-explanatory and does not require the adjective "extraordinary".

We have removed the word “extraordinary” as requested.

2) The truncation in Eq. (2) may not be valid on resonance.

Truncation of Eq. (2) after the AOA term is done assuming a weakly polarizable object. This covers molecular vibrations, but indeed does not cover resonant materials such as phononic materials. We discuss this limitation of our model in Extended Data Fig. 1.

3) page 11. I do not understand the sentence:

Scanning the reference arm mirror (variation of E^{ref}) thus allowed us to suppress the interference term in the scattered field intensity, $2\text{Re}\{E^{\text{ref}*} \cdot E_{\text{AOA}}\}$ (Eq. (10)), and to measure directly E^{sca} in amplitude and phase. To further suppress the direct antenna scattering,

I expected to read $E^{\text{ref}} \cdot E_{\text{AOA}}$.

Scanning the mirror enables to perform the spectral analysis but cannot suppress the antenna contribution. Modulating the object-antenna distance enables to suppress the antenna direct contribution.

The reviewer is correct, we have not explained this situation clearly. We have moved this discussion to the Methods section and provide a more detailed explanation:

Page 26

Specifically, the antenna scattered field \mathbf{E}^{sca} was superposed with the external reference field of nano-FTIR interferometer \mathbf{E}^{ref} , yielding a detector signal proportional to $I_d \propto |\mathbf{E}^{\text{sca}}|^2 + |\mathbf{E}^{\text{ref}}|^2 + 2\text{Re}\{\mathbf{E}^{\text{ref}*} \cdot \mathbf{E}^{\text{sca}}\}$. With Eqs. (3),(4), the detector signal can be approximated as

$$I_d \propto |\mathbf{E}_A|^2 + 2\text{Re}\{\mathbf{E}_A^* \cdot \mathbf{E}_{\text{AOA}}\} + 2\text{Re}\{\mathbf{E}^{\text{ref}*} \cdot \mathbf{E}_A\} + 2\text{Re}\{\mathbf{E}^{\text{ref}*} \cdot \mathbf{E}_{\text{AOA}}\} + |\mathbf{E}^{\text{ref}}|^2. \quad (1)$$

To suppress the direct antenna scattering, \mathbf{E}_A , in Eq. (1), the near-field probe was vertically vibrated sinusoidally with amplitude $\Delta d \sim 100$ nm and frequency $\Omega = 256$ kHz, $d = d_0 + \Delta d \cos \Omega t$. Demodulation of the detector signal at a frequency $n\Omega$ ($n = 3$) then suppresses terms $|\mathbf{E}_A|^2$, $2\text{Re}\{\mathbf{E}^{\text{ref}*} \cdot \mathbf{E}_A\}$ and $|\mathbf{E}^{\text{ref}}|^2$ in Eq. (1). However, demodulation alone cannot suppress the interference of field-enhanced molecular scattering, \mathbf{E}_{AOA} , with the direct antenna scattering, \mathbf{E}_A (term $2\text{Re}\{\mathbf{E}_A^* \cdot \mathbf{E}_{\text{AOA}}\}$ in Eq. (1)). To suppress this term and isolate $2\text{Re}\{\mathbf{E}^{\text{ref}*} \cdot \mathbf{E}_{\text{AOA}}\}$, it is further necessary to modulate the phase of \mathbf{E}^{ref} by translating the reference arm mirror (analogous to s-SNOM where metal tips are used to measure the complex refractive index of a sample)³². Note that translation of the reference arm mirror also provides for spectral analysis of \mathbf{E}_{AOA} following the Fourier-transform approach: [...].

4) This observation provides clear evidence that the interaction between antenna and molecular vibration is a scattering process, and that field-enhanced molecular scattering can be measured and is a significant quantity.

Any far-field measurement is a measurement of a scattered field.

We agree. Nevertheless, SEIRA is typically not understood along these lines, and we hope that our manuscript can provide and highlight this clarification.

5) *page 14:*
for all three antennopas.

We corrected this typo, thank you for pointing this out.

REVIEWERS' COMMENTS

Reviewer #2 (Remarks to the Author):

The second revision has improved the paper a lot. I think it is now ready for publication nevertheless several tiny things might be improved, as always in case of complex work.

Reviewer #3 (Remarks to the Author):

The manuscript is definitely improved for a broader audience, as required for Nature Communication. Unfortunately, we regret to say that the rationale behind our recommendation that not to publish in Nature Communications does not change. Please see detailed explanation.

It is encouraging to observe that the authors have acknowledged the previous ambiguities in their manuscript and have made an effort to clarify the assumptions underlying their research. Their willingness to elaborate and provide detailed explanations in response to the review process is a positive development, compared than labelling differing scientific views as “unacceptable” or “incredibly false” in the previous reply, which is not a professional approach. However, it has taken considerable effort through two rounds of reviews to reach this point, underscoring the challenges in ensuring the manuscript is accessible to an audience with diverse backgrounds.

Furthermore, as we pointed out in the previous round of reviewing, from the perspective of a scientist, adhering to fundamental principles of scholarly work, such as accurate interpretation and citation of literature, is essential. Ignoring, misquoting or misrepresenting sources not only diminishes the author’s argument but also jeopardizes the scientific discourse’s integrity. A thorough review of all the citations in the manuscript is recommended to ensure their correct and appropriate usage. It is disappointing to note that this critical feedback was not addressed despite two rounds of reviews.

Reviewer #4 (Remarks to the Author):

The authors have modified the manuscript.

I recommend publication in the present form.

Response to Reviewer 2:

We thank the reviewer for the positive evaluation and the recommendation to accept our manuscript.

The second revision has improved the paper a lot. I think it is now ready for publication nevertheless several tiny things might be improved, as always in case of complex work.

In response to this comment, we felt motivated to re-read our manuscript text carefully once more. We have improved the readability of our manuscript. Particularly, we have clarified and improved on the following aspects:

- We moved details on the numerical calculations (shown in Fig. 1) to the beginning of the theory section to improve understanding of the theory section (Scattering Model)
- We have streamlined the description of the experiment (Experimental verification of [...])
- We have written more clearly the caption to Figs. 3 and 5
- We have written the text describing Fig. 4 more concisely
- We now highlight better the relevance of our experimental study in the discussion section
- Generally, we are now using more consistent wording throughout the manuscript.

Importantly, results and their interpretation are the same and were not changed.

Response to Reviewer 3:

We thank the reviewer for the time to evaluate our manuscript and input to improve our manuscript.

The manuscript is definitely improved for a broader audience, as required for Nature Communication. Unfortunately, we regret to say that the rationale behind our recommendation that not to publish in Nature Communications does not change. Please see detailed explanation.

It is encouraging to observe that the authors have acknowledged the previous ambiguities in their manuscript and have made an effort to clarify the assumptions underlying their research. Their willingness to elaborate and provide detailed explanations in response to the review process is a positive development, compared than labelling differing scientific views as “unacceptable” or “incredibly false” in the previous reply, which is not a professional approach. However, it has taken considerable effort through two rounds of reviews to reach this point, underscoring the challenges in ensuring the manuscript is accessible to an audience with diverse backgrounds.

Furthermore, as we pointed out in the previous round of reviewing, from the perspective of a scientist, adhering to fundamental principles of scholarly work, such as accurate interpretation and citation of literature, is essential. Ignoring, misquoting or misrepresenting sources not only diminishes the author’s argument but also jeopardizes the scientific discourse’s integrity. A thorough review of all the citations in the manuscript is recommended to ensure their correct and appropriate usage. It is disappointing to note that this critical feedback was not addressed despite two rounds of reviews.

We have reviewed the citations in the manuscript and found them to be correct and appropriate. The references suggested by the reviewer in the last iteration are about Raman spectroscopy and do not apply to our work, which is about SEIRA. In this iteration, the reviewer does not suggest any specific references and thus no further references have been added this time.

Response to Reviewer 4:

The authors have modified the manuscript.
I recommend publication in the present form.

We thank the reviewer for the positive evaluation and recommendation to publish our manuscript.